# Estimation of surface energy fluxes in the Arctic tundra using the remote sensing thermal-based Two-Source Energy Balance model

Jordi Cristóbal[1,2], Anupma Prakash[1], Martha C. Anderson[3], William P. Kustas[3], Eugénie S. Euskirchen[4], Douglas L. Kane[2]

[1]Geophysical Institute. University of Alaska Fairbanks, Fairbanks, Alaska, 99775, USA
[2]Institute of Northern Engineering. Water Environmental Research Center, University of Alaska Fairbanks, Fairbanks, Alaska, 99775, USA
[3]Hydrology and Remote Sensing Laboratory, United States Department of Agriculture, Agriculture Research Service, Beltsville, Maryland, 20705, USA
[4]Institute of Arctic Biology. University of Alaska Fairbanks, Fairbanks, Alaska, 99775, USA

*Correspondence to*: Jordi Cristóbal (j.cristobal@alaska.edu)

**Abstract.** The Arctic has become generally a warmer place over the past decades leading to earlier snow melt, permafrost degradation and changing plant communities. Increases in precipitation and local evaporation in the Arctic, known as the acceleration components of the hydrologic cycle, coupled with land cover changes, have resulted in significant changes in

the regional surface energy budget. Quantifying spatiotemporal trends in surface energy flux partitioning is a key to forecasting ecological responses to changing climate conditions in the Arctic. An extensive local evaluation of the two-source energy balance model (TSEB) - a remote sensing-based model using thermal infrared retrievals of land-surface temperature - was performed using tower measurements collected over different tundra types in Alaska in all sky conditions over the full growing season from 2008 to 2012. Based on comparisons with flux tower observations, refinements in the

original TSEB net radiation, soil heat flux and canopy transpiration parameterizations were identified for Arctic tundra. In particular, a revised method for estimating soil heat flux based on relationships with soil temperature was developed, resulting in significantly improved performance. These refinements result in mean turbulent flux errors generally less than $50 \, \text{W} \cdot \text{m}^{-2}$ at half-hourly timesteps, similar to errors typically reported in surface energy balance modelling studies conducted in more temperate climatic regimes. The MODIS leaf area index (LAI) remote sensing product proved to be useful for

estimating energy fluxes in Arctic tundra in the absence of field data on local biomass amount. Model refinements found in this work at the local scale build toward a regional implementation of the TSEB model over Arctic tundra ecosystems, using thermal satellite remote sensing to assess response of surface fluxes to changing vegetation and climate conditions.

## 1 Introduction

Air temperatures in the Alaskan Arctic have shown a significant increase, especially in past decade (Serreze and Barry,

2011). Results from models forced with a range of climate scenarios from the Intergovernmental Panel on Climate Change (IPCC) indicate that by the mid-21st century the permafrost area in the Northern Hemisphere is likely to decrease by 37–

81% ( IPCC, 2014). In general, the Arctic has become a warmer place, leading to an acceleration of the hydrologic cycle, earlier snow melt, and drier soils due to permafrost degradation (AMAP, 2012; Elmendorf et al., 2012; Rawlins et al., 2010; Sturm et al., 2001; Overduin and Kane, 2006). Furthermore, the hydrologic response of the Arctic land surface to changing climate is dynamically coupled to the region's surface energy balance (Vörösmarty et al., 2001), and the partitioning of

energy fluxes plays an important role in modulating the hydrologic cycle of Arctic basins (Rawlins et al., 2010).

Evapotranspiration (ET, in units of mass, kg s$^{-1}$ m$^{-2}$ or mm d$^{-1}$) or equivalently, latent heat flux (LE, in energy units, W·m$^{-2}$), is an important component of both the land surface hydrologic cycle and surface energy balance. As an example, Kane et al. (2004) reported water loss due to ET in the Imnavait Creek Basin in Alaska is about 74% of summer precipitation or 50% of annual precipitation, as estimated from water balance computations. Even though ET is a significant component of the

hydrologic cycle in Arctic regions, it is poorly quantified in Arctic basins, and the bulk estimates do not accurately account for spatial and temporal variability due to vegetation type and topography (Kane and Yang, 2004). In the Arctic, values of ET or LE are usually either derived from field estimates (Kane et al., 1990; Mendez et al., 1998) or calculated purely from empirical or quasi-physical models such as those described by Zhang et al. (2000) and Shutov et al. (2006) using meteorological station forcing data. However, due to remoteness, harsh winter conditions and the high costs of maintaining

ground-based measurement networks, the data currently collected are also both temporally and spatially sparse.

In Arctic tundra ecosystems, several factors have contributed to the vegetation change such as increased extent of severe fires, increased extent in deciduous vegetation or shrub encroachment in tundra ecosystems (Myers-Smith et al., 2011; Sturm et al., 2001), among others. Over at least the past three decades, Arctic ecosystems have shown evidence of "greening" (Xu et al., 2013; Bhatt et al., 2010), with about a 17% increase in peak vegetation greenness for the Arctic tundra biome (Jia et

al., 2003). Moreover, the forest-tundra transition zone is observed to be moving further north, tree heights are increasing, and shrubs are becoming denser and taller (ACIA, 2004; AMAP, 2012). These changes in vegetation will have an important impact on the surface energy balance, especially in areas where shrubs have made their appearance in former tundra vegetation. This increase in leaf area index (LAI), together with canopy height, and changes in the distribution of canopy elements, will augment the multiple scattering and absorption of radiation, likely resulting in a lower albedo (Beringer et al.,

2005), although more detailed observations and measurements, particularly for the beginning of the snow-free period and peak growing season are needed (Williamson et al., 2016). Also, according to Beringer et al. (2005), Bowen ratio increases from tundra to forested sites will result in an increasing dominance of sensible heat (H) as the primary energy source heating the atmosphere. In the case of a transition from tundra to tall shrub and then to forest, H would likely increase during the growing season from ~15% to nearly 30%, respectively. This will have an important impact in the tundra energy

partitioning, resulting in a positive feedback to the atmosphere that further warms the Arctic climate. However, the magnitude of changes in surface energy partitioning due to vegetation changes and resulting impact on local Arctic climate is still unclear and more research is needed to better understand these vegetation change-atmosphere dynamics (Eugster et al., 2000; Jung et al., 2010).

In the last two decades, surface energy balance methods have demonstrated their utility in modelling water availability using diagnostic retrievals of energy fluxes from *in situ* or remote sensing data, especially data acquired in the thermal infrared (TIR) region (Kalma et al., 2008). While remote sensing estimates of ET over the Arctic exist from global modelling systems (Mu et al., 2009; Zhang et al., 2010), these modelling systems typically do not compute the full energy balance. To estimate

energy fluxes at local scales, on the order of hundreds of meters, initiatives such as FLUXNET (http://fluxnet.ornl.gov/) provide eddy covariance flux measurements at discrete sites situated in different ecosystems across the U.S. and globally. Unfortunately, there are few measurements sites in the Arctic (Mu et al., 2009), making the existing instrument network insufficient to capture pertinent details of the changing Arctic climate and landscape (ACIA, 2004; AMAP, 2012; Serreze and Barry, 2011; Vörösmarty et al., 2001). Detailed process-based (prognostic) land-surface models can be also used to

estimate coupled water and energy fluxes over landscapes (Duursma and Medlyn, 2012; Ek et al., 2003; Falge et al., 2005; Haverd et al., 2013; Smith et al., 2001; Vinukollu et al., 2012, among others); however, they may neglect important processes that are not known a priori. For example, Hain et al. (2015) demonstrated the value of comparing prognostic and TIR-based diagnostic latent heat flux estimates over the continental U.S. to diagnose moisture sources and sinks that were not well-represented in the prognostic modelling system.

Given the critical need to better understand the water and energy balance over tundra ecosystems, and the role of changing climate and vegetation cover in driving these budgets, the aim of this work is to evaluate and refine a diagnostic TIR remote sensing-based model for estimating seasonal dynamics of surface energy fluxes (LE, H, net radiation ($R_N$) and soil heat flux (G)) as well as energy partitioning in the Arctic tundra growing season from 2008 to 2012. Specifically, a refined version of the Two-Source Energy Balance model (TSEB, Norman et al. (1995)) for Arctic tundra is evaluated with in situ forcing data

from three eddy covariance flux towers in all sky conditions, and remote sensing estimates of vegetation properties (LAI, NDVI (normalized difference vegetation index) and EVI (enhanced vegetation index)) from the Moderate Resolution Imaging Spectroradiometer (MODIS). Model refinements include a new G parameterization and new configurations to retrieve $R_N$ (effective atmospheric emissivity), H and LE (two different Priestley-Taylor configurations). The TSEB serves as the land surface scheme in a regional Atmosphere-Land Exchange Inverse (ALEXI) modelling system (Anderson et al.,

2011), implemented operationally over North America as part of NOAA's GOES Evapotranspiration and Drought Information System. Although the TSEB has been demonstrated to work well over a range in vegetation and climate conditions at mid-latitudes (Anderson et al., 2007, 2011; Choi et al., 2009; Sánchez et al., 2009; Tang, et al., 2011; Timmermans et al., 2007, among others), it has not yet been examined for tundra ecosystems characteristic of high latitudes.

## 2 Two-Source Energy Balance model: an overview

Evapotranspiration (ET) can be estimated by surface energy balance models that partition the energy available at the land surface ($R_N$ - G, where $R_N$ is net radiation and G is the soil heat flux, both in W·m$^{-2}$) into turbulent fluxes of sensible and latent heating (H and LE, respectively, in W·m$^{-2}$):

$$LE + H = R_N - G, \tag{1}$$

where L is the latent heat of vaporization (J kg$^{-1}$) and E is ET (kg s$^{-1}$ m$^{-2}$ or mm s$^{-1}$).

The model used in this study is the series version of the Two-Source Energy Balance (TSEB) scheme originally proposed by Norman et al. (1995), which has been revised to improve shortwave and longwave radiation exchange within the soil–canopy system and the soil–canopy energy exchange (Kustas and Norman, 1999, 2000). A list of the TSEB inputs can be found in Table 1. TSEB has been successfully applied over rain fed and irrigated crops and grasslands in temperate and semi-arid climates (Anderson et al., 2012; Anderson et al., 2004; Cammalleri et al., 2012, 2010) but has not been previously applied over the Arctic tundra.

In the TSEB, directional surface radiometric temperature derived from satellite or a ground-based radiometer, $T_{RAD}(\theta)$ (K), is considered to be a composite of the soil and canopy temperatures, expressed as:

$$T_{RAD}(\theta) \approx [f_c(\theta)T_c^4 + (1 - f_c(\theta))T_s^4]^{1/4}, \tag{2}$$

where $T_C$ is canopy temperature (K), $T_S$ is soil temperature (K), and $f_C(\theta)$ is the fractional vegetation cover observed at the radiometer view angle $\theta$. For a canopy with a spherical leaf angle distribution and LAI, $f_C(\theta)$ can be estimated as:

$$f_c(\theta) = 1 - exp\left(\frac{-0.5\Omega LAI}{cos\theta}\right), \tag{3}$$

where the factor $\Omega$ indicates the degree to which vegetation is clumped as in row crops or sparsely vegetated shrub land canopies (Kustas and Norman, 1999, 2000). The composite soil and canopy temperatures are used to compute the surface energy balance for the canopy and soil components of the combined land-surface system:

$$R_{NS} = H_S + LE_S + G, \tag{4}$$

$$R_{NC} = H_C + LE_C, \tag{5}$$

where $R_{NS}$ is net radiation at the soil surface, $R_{NC}$ is net radiation divergence in the vegetated canopy layer, $H_C$ and $H_S$ are canopy and soil sensible heat flux, respectively, $LE_C$ is the canopy transpiration rate, $LE_S$ is soil evaporation, and G is the soil heat flux. The net shortwave radiation is calculated from the measured incoming solar radiation and the surface albedo, while net longwave radiation is estimated from the observed air and land surface temperatures, using the Stefan-Boltzmann equation with atmospheric emissivity from the Brutsaert (1975) method.

By permitting the soil and vegetated canopy fluxes to interact with each other, Norman et al. (1995) derived expressions for $H_S$ and $H_C$ expressed as a function of temperature differences where:

$$H_S = \rho C_p \frac{T_S - T_{AC}}{r_S}, \tag{6}$$

and

$$H_C = \rho C_p \frac{T_C - T_{AC}}{r_X}, \tag{7}$$

with the total sensible heat flux $H = H_C + H_S$ expressed as:

$$H = \rho C_p \frac{T_{AC} - T_A}{r_A}, \tag{8}$$

where $\rho$ is air density (kg·m$^{-3}$), $C_p$ is the specific heat of air (kJ·kg$^{-1}$·K$^{-1}$), $T_{AC}$ is air temperature in the canopy air layer (K), $T_A$ is the air temperature in the surface layer measured at some height above the canopy (K), $r_X$ is the total boundary layer resistance of the complete canopy of leaves (s m$^{-1}$), $r_S$ is the resistance to sensible heat exchange from the soil surface (s m$^{-1}$) and $r_A$ is aerodynamic resistance (s m$^{-1}$) defined by:

$$R_A = \frac{[ln((z_U-d_O)/z_{OM})-\Psi_M][ln((z_T-d_O)/z_{OM})-\Psi_H]}{k^2 u}, \tag{9}$$

In Eq. (9) $d_O$ is the displacement height, u is the wind speed measured at height $z_U$, k is von Karman's constant ($\approx 0.4$), $z_T$ is the height of the $T_A$ measurement, $\Psi_M$ and $\Psi_H$ are the Monin–Obukhov stability functions for momentum and heat, respectively, and $z_{OM}$ is the aerodynamic roughness length.

The original resistance formulations are described in more detail in Norman et al. (1995) with revisions described in Kustas and Norman (1999) and Kustas and Norman (2000). Weighting of the heat flux contributions from the canopy and soil components is performed indirectly by the partitioning of the $R_N$ between soil and canopy and via the impact on resistance values from the fractional amount and type of canopy cover (see Kustas and Norman, 1999).

For the latent heat flux from the canopy, the Priestley–Taylor formula is used to initially estimate a potential rate for $LE_C$:

$$LE_C = \alpha_{PTC} f_G \frac{\Delta}{\Delta+\gamma} R_{NC}, \tag{10}$$

where $\alpha_{PTC}$ is a variable quantity related to the Priestley–Taylor coefficient (Priestley and Taylor, 1972), but in this case defined exclusively for the canopy component, which was suggested for row crops by Tanner and Jury (1976) and normally set to an initial value of 1.2, $\Delta$ is the slope of the saturation vapour pressure versus temperature curve and $\gamma$ is the psychrometric constant ($\sim$0.066 kPa °C$^{-1}$). $f_G$ is the fraction of green vegetation that according to Guzinski et al. (2013) and Fisher et al., (2008) can be estimated through the normalized difference vegetation index (NDVI) and the enhanced vegetation index (EVI):

$$f_G = 1.2 \frac{EVI}{NDVI}, 0 \leq f_G \leq 1, \tag{11}$$

Under stress conditions, TSEB iteratively reduces $\alpha_{PTC}$ from its initial value. The TSEB model requires both a solution to the radiative temperature partitioning (Eq. 2) and the energy balance (Eqs. 6 and 7), with physically plausible model solutions for soil and vegetation temperatures and fluxes. Non-physical solutions, such as daytime condensation at the soil surface (i.e., $LE_S < 0$), can be obtained under conditions of moisture deficiency. This happens because $LE_C$ is overestimated in these cases by the Priestley–Taylor parameterization, which describes potential transpiration. The higher $LE_C$ leads to a cooler $T_C$ and $T_S$ must be accordingly larger to satisfy Eq. (7). This drives $H_S$ higher, and the residual $LE_S$ from Eq. (12) can become negative. If this condition is encountered by the TSEB scheme, αPTC is iteratively reduced until $LE_S \sim 0$ (expected for a dry soil/substrate surface). However there are instances where the vegetation is not transpiring at the potential rate but is not stressed due to its adaption to water and climate conditions (Agam et al., 2010) or the fact that not all the vegetation is green

or actively transpiring (Guzinski et al., 2013) (a thorough discussion of conditions that force a reduction in $\alpha_{PTC}$, can be also found in Anderson et al. (2005) and Li et al. (2005)).

The latent heat flux from the soil surface is solved as a residual in the energy balance equation:

$$LE_S = R_{NS} - G - H_S, \tag{12}$$

with G estimated as a fraction of the net radiation at the soil surface ($c_G$):

$$G = c_G R_{NS}, \tag{13}$$

From midmorning to midday period the value of $c_G$ can be typically assumed to be constant (Kustas and Daughtry, 1990;Santanello and Friedl, 2003). In this case, a typical value of ~0.3 can be assumed for $c_G$ based on experimental data from several sources (Daughtry et al., 1990). However, $c_G$ value varies with soil type and moisture conditions as well as

time, due to the phase shift between G and $R_{NS}$ over a diurnal cycle (Santanello and Friedl, 2003).

## 3 TSEB formulation refinements for Arctic tundra

### 3.1 Downwelling longwave radiation estimation: effective atmospheric emissivity for all sky conditions

The original TSEB formulation estimates the downwelling longwave radiation component of $R_N$ using the effective atmospheric emissivity ($\varepsilon$) method described in Brutsaert (1975) for clear sky conditions:

$$\varepsilon = C(e/T_A)^{1/7}, \tag{14}$$

where e is the water pressure in millibars and $T_A$ in K and C is 1.24 as in the original Brutsaert (1975) formulation. However, in this study TSEB is applied for all sky conditions, including clear sky, partially cloudy and overcast conditions. To estimate $\varepsilon$ for all sky conditions Crawford and Duchon (1999) proposed a methodology that incorporated the Brutsaert (1975) clear-sky parameterization and the Deardorff (1978) cloudiness correction using a simple cloud modification

introducing a cloud fraction term (clf) according to the following equation:

$$\varepsilon = \{clf + (1 - clf)[C(e/T_A)^{1/7}]\}, \tag{15}$$

The clf is defined as:

$$clf = 1 - s, \tag{16}$$

where s is the ratio of the measured solar irradiance to the clear-sky irradiance. Shortwave clear-sky irradiance used in Eq.

(16) may be obtained through the methodology proposed by Pons and Ninyerola (2008), where incident clear-sky irradiance is calculated through a digital elevation model at a specific point during a particular day of the year taking into account the position of the Sun, the angles of incidence, the projected shadows, the atmospheric extinction and the distance from the Earth to the Sun.

For Arctic areas, Jin et al. (2006) suggested an improved formulation of C for clear sky conditions that can also be applied in

Eq. (15) for all sky conditions, defined as:

$$C = 0.0003(T_A - 273.16)^2 - 0.0079(T_A - 273.16) + 1.2983, \tag{17}$$

In order to evaluate if the Jin et al. (2006) method offered more accurate estimates of ε for Arctic conditions, this method was compared to Brutsaert (1975) formulation used in TSEB, in both cases for all sky conditions using Eq. (15).

## 3.2 Refinements in soil heat flux parameterization: $c_G$ coefficient and definition of a new coefficient based on $T_{RAD}$

In the Arctic tundra the propagation of the thawing front in the soil active layer consumes a large proportion (around 18%) of the energy input from the positive net radiation (Boike et al., 2008a; Rouse, 1985). Moreover, the presence of permafrost in tundra areas may contribute to the large tundra soil heat flux by creating a strong thermal gradient between the ground surface and depth, offsetting the influence of the highly insulative moss cover which would otherwise have been expected to reduce soil heat flux (Myers-Smith et al., 2011; Sturm et al., 2001). Therefore, previous formulations of soil heat flux used in TSEB applications, mainly representative of cropped and sparse-vegetated areas in the U.S., need to be adjusted and validated for Arctic tundra.

Currently there are several methodologies that allow estimating soil heat flux from tenths of centimetres to meters in depth in the Arctic tundra by using modelling or instrumentation at several depths (Lynch et al., 1999; Ekici et al., 2015; Jiang et al., 2015; Romanovsky et al., 1997; Yao et al., 2011; Zhuang et al., 2001; Hinzman et al., 1998). However, in this study a simple approach based on the relationship between G and $R_{NS}$ (Eq. (13)) was used to estimate the soil heat flux in the near-surface soil layer (around 10 cm depth). This approach has less complexity and requires less input data than the methods mentioned above and allows estimating G at regional scales.

In early TSEB implementation, a constant value of $c_G$ value around 0.3 was used to estimate G for the midmorning to midday period (Eq. (13) based on findings by Kustas and Daughtry (1990) for U.S. study sites. . However, this assumption can result in significant errors if applied out of this time range. For diurnal hourly timescales, Kustas et al. (1998), developed a method to estimate $c_G$ based on time differences with the local solar noon quantified by a non-dimensional time parameter. Although this approach does not consider the phase shift between G and $R_{NS}$ over a diurnal cycle, a phase shift was included in the model proposed by Santanello and Friedl (2003) in the following form:

$$c_G = A \cos[2\pi(t + S)/B],$$
(18)

where $A$ represents the maximum value of $c_G$, $B$ is chosen to minimize the deviation of $c_G$ from Eq. (13), $t$ is time in seconds relative to solar noon and S is the phase shift between G and $R_{NS}$ in seconds. Values fitted for A, S and B were 0.31, 10 800 and 74 000, respectively.

Although $c_G$ values for Arctic tundra were not found in the literature, several studies present (Beringer et al., 2005; Eugster et al., 2005; Boike et al., 2008b; Eaton et al., 2001; Eugster et al., 2000; Kodama et al., 2007; Langer et al., 2011; Soegaard et al., 2001; Westermann et al., 2009; Mendez et al., 1998; Lund et al., 2014) the relationship between $R_{NS}$ and G during the summer months in similar tundra areas. According to these studies, a mean value of 0.14, as a maximum value of $c_G$ in Eq. (18), can be derived from different analyses of $R_{NS}$ and G over the Arctic tundra.

An alternative parameterization for G suggested by Santanello and Friedl (2003) for several types of soils with crops, and by Jacobsen and Hansen (1999) for Arctic tundra that links the soil heat flux to the diurnal variations in surface radiometric temperature. This approach can also be applied for Arctic tundra as follows:

$$G = c_{TG}T_{RAD},$$ (19)

5 where $c_{TG}$ is a coefficient that represents the relationship between the diurnal variation of $T_{RAD}$ and G. For diurnal hourly timescales, $c_{TG}$ can be also estimated using the phase shift proposed in Eq. (18); where in this case, S is the phase shift between G and $T_{RAD}$ in seconds. This new approach avoids using $R_{NS}$, which is more difficult to define in tundra systems given the influence of the surface moss layer above the mineral soil. Moreover, A, S and B in Eq. (18) can be fitted by using direct measurements of $T_{RAD}$ from thermal field sensors, commonly available on flux towers (pyrgeometer), or thermal data 10 from geostationary or polar satellites.

Thus, to evaluate soil heat flux for diurnal hourly timescales, the approaches of Kustas et al. (1998) and Santanello and Friedl (2003) were compared using the original $c_G$ value of 0.30 and a new value for Arctic tundra of 0.14, both as maximum values of $c_G$ in Eq. (18). A, B and S values for the new $c_{TG}$ approach were fitted and tested using an extended evaluation dataset and then compared to these radiation-based methods (see section 4.2).

15 **3.3 Priestley-Taylor coefficient**

In the original TSEB formulation, the Priestley-Taylor approach for the canopy component of LE is used. In this case $\alpha_{PTC}$ is normally set to an initial value of 1.26 for the general conditions tested during the growing season in rangelands and croplands. For stressed canopies, TSEB internally modifies $\alpha_{PTC}$ to yield reasonable partitioning between $LE_C$ and $LE_S$.

As with the $c_G$ coefficient, specific $\alpha_{PTC}$ values for tundra were not found in the literature. Alternatively, measurements of 20 bulk (soil+canopy) for Arctic tundra systems are available (Beringer et al., 2005;Eaton et al., 2001;Eugster et al., 2005;Engstrom et al., 2002;Mendez et al., 1998;Lund et al., 2014) suggesting a mean value of around 0.92. This bulk value might suggest that $\alpha_{PTC}$ could also be lower for summer Alaska tundra conditions. For natural vegetation, Agam et al. (2010) also suggested that a lower $\alpha_{PTC}$ value might yield better results. Therefore, for modelling purposes two different values of $\alpha_{PTC}$ values, 0.92 and 1.26, were applied to evaluate which nominal $\alpha_{PTC}$ input to TSEB was more appropriate for Arctic 25 tundra.

**4 Study area and data description**

**4.1 Study area**

To refine and evaluate the TSEB model for Alaska's Arctic tundra summer conditions, three eddy covariance flux towers (referred to as Fen, Tussock and Heath; see Fig. 1) were selected. These are located across the Imnavait Watershed (~904 m

a.s.l.) with eddy covariance and associated meteorological data collection beginning in 2007 (Euskirchen et al., 2012; Kade et al., 2012). A brief description of instrumentation at the tower sites is provided in Table 2.

The Fen tower, located at the valley bottom in a wet sedge ecosystem, includes *Eriophorum angustifolium* and dwarf shrubs such as *Betula nana* and *Salix* spp and vegetation types around the tower are comprised of 52% wet sedge, and 47% tussock

tundra. The Tussock tower, located at the midslope in a moist acidic tussock tundra ecosystem, is dominated by the tussock-forming sedge *Eriophorum vaginatum*, *Sphagnum* spp., and dwarf shrubs such as *Betula nana* and *Salix* spp. In this case, vegetation types around the flux tower are 95% tussock tundra. The Heath tower sits atop a broad dry ridge at the top edge of the eastern watershed boundary in a heath tundra ecosystem dominated by dwarf shrubs and lichen. The vegetation here is 20% heath, but also included 72% tussock tundra, with the balance made of up of sedge meadow and bare soil. Further

detailed information about the study is provided in Euskirchen et al. (2012) and Trochim et al. (2015).

## 4.2 Model inputs, evaluation datasets, and metrics

### 4.2.1 Micrometeorological input data

Data incorporated in this study spanned from May to September 2008 to 2012. These included eddy covariance data for latent and sensible collected at 10 Hz and processed to 30-minute means (described below) as well as meteorological data

collected at 30-minute intervals (Table 1 and Table 2). These data, from under all sky conditions, were used to refine and evaluate the model performance (Table 1). This dataset was considered to be representative of the short Arctic tundra vegetative cycle from early growing to senescence as well as to capture inter- and intra-annual vegetation dynamics.

Meteorological input for TSEB include wind speed, air temperature, vapour pressure, atmospheric pressure, longwave incoming radiation and solar radiation, all of which were collected at the three measurement sites (see Table 1 and 2). The

surface radiometric temperature $T_{RAD}$ inputs were obtained from the pyrgeometer sensor at the Tussock station and from infrared radiometer sensors at both Fen and Heath stations.

### 4.2.2 Remote sensing input data: vegetation properties

In addition, TSEB also requires estimates of LAI and the fraction of vegetation that is green to specify $f_C$ in Eq. (2) and to estimate $LE_C$ in Eq. (10). While, *in situ* measurements of LAI were not available at the tower sites for the length of this

study, the 500 m combined Terra/Aqua MODIS 4-day LAI product (MCD15A3H) was available for the study area. This product has been successfully applied in other applications of the TSEB (Guzinski et al., 2013) where sites are considered homogeneous over several kilometres, and serve here as a proxy for local observations. The fraction of vegetation that is green ($f_G$) in Eq. (10) was estimated using NDVI and EVI from MODIS imagery using the daily 250 m reflectance product (MOD09GQ), and using the blue band in the daily 500 m reflectance product 500 m (MOD09GA) to correct for residual

atmospheric effects, with negligible spatial artifacts. Because of MODIS time series contains occasional lower quality data, gaps from persistent clouds, cloud contamination, and other gaps (Gao et al., 2008), a program for analysing time series of

remote sensing imagery, TIMESAT (Jönsson and Eklundh, 2004) was used to produce temporally smoothed NDVI, EVI and LAI by selecting the best estimates through these products quality flags. Gao et al. (2008) found a good agreement with field measurements when smoothing MODIS LAI data using this distribution and several weights (w) based on the product quality flags (w = 1.0 for LAI retrievals from the radiative-transfer model (high quality) or for LAI retrieval that reaches

saturation, w = 0.25 for retrievals from an empirical model and w = 0.0 for all invalid and fill values). Beck et al. (2006) also reported that an asymmetric Gaussian distribution was appropriate for describing vegetation dynamics using NDVI at high latitudes and several weights (w) based on the product quality flags (highest quality/clear, mixed and cloudy were assigned weights of 1, 0.5 and 0, respectively). For NDVI, EVI and LAI time series smoothing, the weights and quality flags proposed by Beck et al. (2006) and Gao et al. (2008) were used.

Vegetation height, used to define roughness parameters $d_O$ and $z_{OM}$, was assigned based on measurements made in the vicinity of the flux towers (Kade et al., 2012) and the clumping factor was set to 1 for all sites based on the knowledge that Arctic tundra has a variable moss layer with little bare ground. Variability regarding these inputs for the studied periods is shown in Table 1. Moreover, to ensure that only snow-free periods were analysed, Terra/Aqua MODIS snow cover products (MOD10A1 and MYD10A1) were used to screen days with snow cover at the beginning and end of the growing season.

### 4.2.3 Micrometeorological flux data for model evaluation

The eddy covariance data used in TSEB evaluation, including latent and sensible heat, were processed with EddyPro® (2004) software. Changes in mass flow caused by changes in air density (Webb et al., 1980), corrections for frequency attenuation of eddy covariance fluxes following Massman (2000) and Rannik (2001) and storage corrections for calm

periods (friction velocity (u*) was less than 0.1 m s$^{-1}$ suggested by Rocha and Shaver (2011)) were accounted for. The automatic gain control (AGC) value (which represents optical impedance by precipitation) was computed for the IRGA and used as a QA/QC variable for both flux and radiation data, with 60 as the maximum threshold value (LI-COR 2004). Rejection angles of 10° were also used when the eddy covariance instruments were downwind of a tower to remove flow distortions. In addition, corrections for stationarity, lags, step-change, among others, were performed by the flux processing

software (for further information on micrometeorological data processing see Euskirchen et al. (2012) and http://aon.iab.uaf.edu/data_info). To select the best data available, the above criteria were used to flag the micrometeorological dataset, and no gap-filled data were used.

In addition, soil heat flux plate measurements were corrected to account for soil heat storage above the plate according to the calorimetric methodology proposed by Domingo et al. (2000) and Lund et al. (2014) using existing field measurements of

soil bulk density for each site (758 kg·m$^{-3}$, 989 kg·m$^{-3}$ and 1038 kg·m$^{-3}$ for Fen, Tussock and Heath flux stations, respectively), soil moisture from the water content reflectometer and thermocouple averaging soil temperature probes (TCAV) placed at two depths in the soil (see Table 2).

To evaluate the new $c_{TG}$ approach, a total of 41068 half-hourly timesteps of $T_{RAD}$ and G from 4 to 21 hours local solar time were selected (11593, 14454 and 15021 for Fen, Tussock and Heath flux stations, respectively). Coefficients *A*, *B* and *S* were

fitted using 60% of all available data (fitting subset) aggregated in 30 min timesteps for the whole summer period. The remaining 40% of the data were reserved for model testing (test subset) (see Table 4 for flux stations distribution). To evaluate the TSEB model, including G retrieve from Kustas et al. (1998) and Santanello and Friedl (2003) approaches, a total of 5178 half-hourly timesteps (1558, 1273 and 2347 for Fen, Tussock and Heath flux stations, respectively) was subset
from the previous selection by imposing three criteria: a) energy closure at the half-hourly timescale exceeded 70%, b) $R_N$ was higher than 100 W·m$^{-2}$ in order to ensure daylight conditions, and c) no precipitation present.

### 4.2.4 Evaluation metrics

For model evaluation, surface energy fluxes ($R_N$, LE, H and G) from the flux datasets (observed values) were compared to TSEB outputs (estimated values) using five metrics describing model errors and biases: the coefficient of determination ($R^2$)
was used to indicate the precision of the estimates in relation to observed surface energy fluxes; the root mean square error (RMSE) was used as a measure of accuracy to measure differences between values estimated by the TSEB model and values actually observed by the flux towers ; the mean bias error (MBE) was used to indicate cumulative offsets between measured and observed values; the mean absolute difference (MAD) was used to indicate the magnitude of the average absolute difference of observed and estimated values; and finally,  the mean absolute percent difference (MAPD) was used to express
the magnitude of absolute difference between observed and estimated values relative to the observed mean value, from Eq. (20) to Eq. (24), respectively.

$$R^2 = \left( \frac{\sum_{i=1}^{n}(o_i-\bar{O})(e_i-\bar{E})}{\sqrt{\sum_{i=1}^{n}(o_i-\bar{O})^2}\sqrt{\sum_{i=1}^{n}(e_i-\bar{E})^2}} \right), \tag{20}$$

$$RMSE = \sqrt{\frac{\sum_{i=1}^{n}(e_i-o_i)^2}{n}}, \tag{21}$$

$$MBE = \frac{\sum_{i=1}^{n}(e_i-o_i)}{n}, \tag{22}$$

$$MAD = \frac{\sum_{i=1}^{n}|e_i-o_i|}{n}, \tag{23}$$

$$MAPD = \frac{100}{n}\left( \frac{\sum_{i=1}^{n}|e_i-o_i|}{\bar{O}} \right), \tag{24}$$

where $e_i$ refers to the estimated value of the variable in question ($R_N$, H, LE or G), $o_i$ is the observed value (*in situ* measurement provided by the flux station), n is the number of data points, and $\bar{O}$ and $\bar{E}$ are the average of the $o_i$ and $e_i$ values, respectively.

# 5 Results and discussion

## 5.1 Evaluation of soil heat flux model refinements for tundra

Both the Kustas et al., (1998; K98) and the Santanello and Friedl (2003; SF03) soil heat flux models used to estimate G at the study sites yielded high errors when a value of $c_G = 0.3$ was used, with MAPD ranging from 90% to 159%. In this case, the SF03 approach provided better results (Table 3). It is important to note that G is a relatively small term with a maximum value on the order of 50 W·m$^{-2}$. Both models generally overestimated G with a MBE from 3 W·m$^{-2}$ to 40 W·m$^{-2}$, with the SF03 model generating lower biases. Results improved when a $c_G$ value of 0.14 was used with MAPD ranging from 48% to 76% and with lower RMSE values from 15 W·m$^{-2}$ to 21 W·m$^{-2}$ and MBE from -4 W·m$^{-2}$ to -14 W·m$^{-2}$. With the lower value of $c_G$, the K98 approach provided better results (Table 3).

Similar to the original $c_G$, $c_{TG}$ can be also estimated using the Santanello and Friedl (2003) method in Eq. (18). Mean diurnal profiles in $T_{RAD}$ and G, averaged over all tundra (see section 4.2 and Table 3) showed a phase shift between these variables (Fig. 2). The mean G value for the summer period peaked around 15:00 local solar time, with a phase shift around 4 hours after the maximum $T_{RAD}$ at noon. Using $T_{RAD}$ and G observations at half-hourly timesteps from the fitting subset, diurnal $c_{TG}$ curves were derived for the growing season for each of the tower sites, showing reasonable agreement (Fig. 3). A fit to the mean curve yielded parameter values of $S$ = -14 400 seconds, $A$ = 1.55 and $B$ = 160 000 s. As in the case of Santanello and Friedl (2003), a $B$ variation of ± 15 000 s had no significant influence on the results. Statistical comparisons between observed fluxes from the test subset and simulations using the fitted parameters show good agreement and negligible bias (Table 4), with $R^2$, MAPD, RMSE and MBE values of 0.68, 37%, 6 W·m$^{-2}$ and 0 W·m$^{-2}$, respectively. In addition, the new model was also evaluated using the same flux subset used in Table 3 to assess the K98 and SF03 configurations, demonstrating improved performance with roughly half the MAPD than K98 and SF03 configurations (Table 4).

The performance of the G parameterization for Arctic tundra reported here is comparable or superior to previous studies reported in the literature using the Santanello and Friedl (2003) or Kustas et al. (1998) appoaches for other ecosystems. In shrub-grass dominated areas and boreal forest several studies (Anderson et al., 2008; Kustas et al., 1998; Li et al., 2008; Sánchez et al., 2009; Timmermans et al., 2007) reported MAPD and RMSE values ranging from 19% to 59% and from 15 W·m$^{-2}$ to 35 W·m$^{-2}$, respectively. Studies in corn and soybean crops (Anderson et al., 2005; Choi et al., 2009; Li et al., 2005; Santanello and Friedl, 2003) reported MAPD and RMSE values ranging from 19% to 34% and from 10 W·m$^{-2}$ to 41 W·m$^{-2}$, respectively.

## 5.2 Net radiation evaluation: effective atmospheric emissivity

Effective atmospheric emissivity estimated using the Brutsaert (1975) and Jin et al. (2006) methodologies yielded similar errors in simulated downwelling longwave radiation results, with a $R^2$ of 0.58 and a RMSE of 26 W·m$^{-2}$ and 27 W·m$^{-2}$, respectively. The C coefficient computed through Jin et al. (2006) yielded a value of 1.25 ±0.009, very close to Brutsaert (1975) C value of 1.24. This suggests that the simpler Brutsaert (1975) C coefficient can be used efficiently to model

effective atmospheric emissivity in all sky conditions when combined with Crawford and Duchon (1999) and Pons and Ninyerola (2008) methods for summer Arctic tundra.

Estimated $R_N$ for all sky condition yielded strong agreement with observed values for all flux towers (see Fig. 4 and Table 5) with a mean $R^2$, MAPD, MAD, RMSE of 0.99, 7%, 18 W·m$^{-2}$, 23 W·m$^{-2}$, with a tendency to overestimate $R_N$ with a MBE of 7 W·m$^{-2}$. In terms of RMSE and MAPD, all study sites behaved similarly (see Fig. 4). These results are in line with previous TSEB model applications for other cover types and clear sky conditions where a MAPD of around 5% was reported (Anderson et al., 2008; Li et al., 2005; Anderson et al., 2005; Kustas and Norman, 1999; Guzinski et al., 2013; Li et al., 2008; Anderson et al., 2000). This suggest that $R_N$ estimation using this scheme can be applied regionally under summer all sky conditions in Arctic tundra when a source of solar radiation (METEOSAT or GOES, Cristóbal and Anderson (2013)), air temperature (Cristóbal et al., 2008) and $T_{RAD}$ (MODIS Land Surface Temperature and emissivity product) are available.

### 5.3 Latent and sensible heat fluxes evaluation: $\alpha_{PTC}$ configuration for Arctic tundra

The average energy balance closure using half-hour periods for the evaluation dataset was 88% which is in agreement with the average closure of 90% for these flux stations, (Euskirchen et al., 2012). Lack of closure may be explained by instrument and methodological uncertainties, insufficient estimation of storage terms, unmeasured advective fluxes, landscape scale heterogeneity or instrument spatial representativeness, among others (Lund et al., 2014; Stoy et al., 2013; Foken et al., 2011; Foken, 2008; Wilson et al., 2002). More recently, there is evidence that non-orthogonal sonics underestimate vertical velocity causing under-measurement of H and LE on the order of 10% (Kochendorfer et al., 2012; Frank et al., 2013), although this is still being debated (Kochendorfer et al., 2013). While, currently, there is no uniform answer on how to deal with non-closure of the energy balance in eddy covariance datasets and methods for analysing the reasons for the lack of closure are still under discussion (Foken et al., 2011), in this study TSEB output is primarily compared with eddy covariance fluxes as observed, without closure corrections. However, to facilitate comparisons to numerous studies in the literature imposing energy conservation to eddy covariance data when evaluating surface energy balance models (Courault et al., 2005; Kalma et al., 2008), and given that strong evidence is presented in the literature that both H and LE are under-measured by the eddy covariance technique, additional comparisons with closed fluxes using the Bowen ratio ($H_{BR}$ and $LE_{BR}$) approach suggested by Twin et al. (2000) and LE recalculated as the residual ($LE_{RES}$, e.g., Li et al., 2008) are provided for completeness. Results with closed fluxes are presented to provide bounds on the range in probable model performance and to demonstrate the impact of closure corrections on model evaluation metrics.

LE and H estimated through both the new proposed soil heat flux methodology and the all sky $R_N$ methodology scheme, yielded reasonable agreement with observed half-hourly unclosed turbulent fluxes, for both $\alpha_{PTC}$ parameterizations of 0.92 and 1.26 (see Tables 4 and 5, and Fig. 4), although $\alpha_{PTC} = 0.92$ yielded marginally lower errors for H and LE. Relative errors (MAPD) were 40 and 25% for LE and H, respectively, for all combined sites using $\alpha_{PTC} = 0.92$, and 45 and 27% using the

standard value of $\alpha_{PTC} = 1.26$, respectively. A slight improvement in H and LE estimates using $\alpha_{PTC} \sim 0.9$ also agrees with Agam et al. (2010) who also found better results with lower $\alpha_{PTC}$ for natural vegetation in water limited environments.

When energy balance closure is imposed, model performance is mainly improved for LE (up to 10% decrease in MAPD) in
part due to the fact that the measured turbulent fluxes are adjusted to achieve energy conservation as required by surface energy balance models. Indeed, the relatively small errors between modelled and measured $R_N$ (Table 5) and relatively small and unbiased magnitude in modelled/measured G (Table 5 and Figure 4) suggests the unclosed turbulent fluxes contribute to the error statistics comparing Tables 5 and 6 with closed H and LE observed fluxes.

Nevertheless, since the mean RMSE for all fluxes compared to unclosed and closed turbulent fluxes and for all parameterizations and sites was around 50 W·m$^{-2}$ (Table 5 and 6), which is commensurate with errors typically reported in other surface energy balance studies (Kalma et al., 2008), these results suggest that a generalized $\alpha_{PTC}$ value of 1.26 in global TSEB applications may adequately reproduce energy fluxes in Arctic tundra during the growing season, from leaf-out until senescence, while also capturing inter- and intra-annual dynamics. However, biases in regional applications may be reduced
by using a land cover class-dependent value of $\alpha_{PTC}$.

Currently, there is limited research published on application of energy balance models to estimate energy fluxes for Arctic tundra. Mu et al. (2009) reported year-round errors from 20% to 40% in two Arctic tundra sites in Barrow (Alaska, USA) at daily periods based on a modified aerodynamic resistance–surface energy balance model where the required surface
conductance is estimated from remotely-sensed LAI based on Cleugh et al. (2007) formulation. TSEB results, however, were evaluated with half-hourly data in summer conditions and, although they cannot be directly compared with results in this previous study, they show similar errors. As in the case of $R_N$, LE and H results are also in line with previous works for other cover types using *in situ* data as input to TSEB (Anderson et al., 2000; Anderson et al., 2008; Li et al., 2005).

### 5.4 Seasonal dynamics of surface energy fluxes and energy partitioning

In general, monthly estimation of surface energy fluxes showed a good agreement with observations during the growing season. Because the model yielded similar results with both $\alpha_{PTC}$ parameterizations of 0.92 and 1.26, this section only shows the seasonal dynamics with $\alpha_{PTC}$ of 0.92. Because of the under-measurement issues with eddy covariance data and greater uncertainty and error associated with LE measurements (Wolf et al., 2008), seasonal dynamics of turbulent fluxes were compared with residual LE. Estimated $R_N$ yielded a low MAPD around 6%, increasing up to 12% at the end of the growing
season (Table 7 and Fig. 5). The proposed new method to estimate G yielded better MAPD results from June to August which coincides with the peak of the growing season in July. A similar pattern was found for LE and H, where the best MAPD results occurred also in the middle of the growing season (June and July). MAPD for LE, H and G tended to be

higher in May and September; thus coinciding with earlier plant growth or the senesce periods, respectively. MODIS LAI product, used to estimate the fractional vegetation cover (Eq. 3) to partition soil and canopy temperatures, performed as a good proxy to capture inter- and intra-annual vegetation dynamics (Fig. 6). Mean seasonal MODIS LAI from May to September for all flux stations was $1.2 \pm 0.5$ $m^2 \cdot m^{-2}$. In previous studies close to the study area, Toolik Lake, and Imnaviat

Creek (Shaver and Chapin, 1991; Shippert et al., 1995; Williams et al., 2001; Williams et al., 2006) reported LAI field estimates ranging from 0.2 to 1.4 $m^2 \cdot m^{-2}$ for different tundra types around mid-July to mid-August, suggesting LAI overestimation from the MODIS product. Loranty et al. (2010) also reported LAI overestimation when using this product in similar tundra types, finding better agreement using a NDVI-LAI relationship (Shaver et al., 2007; Street et al., 2007), although the nonlinearity in the NDVI-LAI conversion is prone to averaging errors when scaled with remote sensing data

(Stoy et al., 2009). Despite MODIS LAI overestimation, it performed well for the Arctic tundra suggesting utility for regional applications, although LAI-NDVI methods might be considered for future applications.

$f_G$ estimated through NDVI and EVI also captured inter- and intra-annual vegetation dynamics (Fig. 6), with a mean seasonal value from May to September for all flux stations of $0.82 \pm 0.7$. From May to August (from beginning and almost to the end of the growing period), $f_G$ showed a good agreement with LAI dynamics. However, while $f_G$ showed a steady increase at the

beginning of the growing season, it did not follow MODIS LAI dynamics in September. This caused the model to overestimate LE and underestimate H during this time period, degrading agreement with observed data. The underperformance of the $f_G$ methods near the end of the growing season might be related to the presence of a variable moss layer, which can exert strong controls on understory water and heat fluxes in Arctic tundra ecosystems (Blok et al., 2011) and may have masked the actual vegetation dynamics (Fig. 6). Further research is needed to confirm this hypothesis. The

pattern of daily estimated surface energy fluxes also compared well to observed fluxes for all sky conditions. As an example, time series of modelled and measured surface energy fluxes are segmented in Fig. 7 for the Heath flux station, with each diurnal segment representing flux data averaged by hour over 5-day intervals from 2008 to 2012. Observed and estimated $R_N$ exhibited an excellent agreement showing almost the same daily temporal pattern for the full growing season while LE, H and G yielded a good daily agreement being underestimated in May and September, especially in the case of LE.

In terms of observed ($_o$) and estimated ($_e$) mean season energy flux partitioning, $LE_o/R_{No}$, $H_o/R_{No}$ and $G_o/R_{No}$ yielded mean values of 0.55, 0.37 and 0.08, respectively; and $LE_e/RN_e$, $H_e/R_{Ne}$ and $G_e/R_{Ne}$ yielded mean values of 0.58, 0.34 and 0.08, respectively (Fig. 8). Observed and estimated Bowen ratio ($\beta$) yielded mean values of 0.60 and 0.67, respectively. In all cases, observed and estimated results are in line with previous studies for Arctic tundra (Lynch et al., 1999; Eugster et al., 2000). It is worth noting that the difference between observed and estimated values of $LE/R_N$, $H/R_N$ partitions was only

around 3% and for $G/R_N$ was almost negligible. From June to August, mean absolute difference values between observed and estimated values for $LE/R_N$, $H/R_N$ were around 4%, increasing up to 15% in September due to model over and underestimation, while $G/R_N$ difference was only less than 1%.

These results suggest that the model is able to reproduce accurately temporal trends of energy partition in concert with tundra vegetation dynamics in the growing vegetation peak from June to August and could be used to monitor changes in surface energy fluxes concurrently with vegetation dynamics.

## 6 Conclusions and future work

Parameterizations for $R_N$, G and $\alpha_{PTC}$ used in the two-source energy balance model (TSEB) were evaluated and refined for applications in different tundra types in Alaska over the full Arctic tundra growing season. Results showed that TSEB may adequately reproduce energy fluxes in Arctic tundra during the growing season, from leaf-out until senescence. The modified TSEB provided turbulent heat flux estimates with a mean RMSE value on the order of 50 W·m$^{-2}$ in comparison with unclosed eddy covariance measurements of H and LE collected at three flux towers – commensurate with errors typically

reported in other surface energy balance studies. Moreover, as in many other studies using eddy covariance flux data for evaluating model performance, imposing energy balance closure yielded better agreement between measured and modelled turbulent fluxes. The all-sky $R_N$ estimation scheme tested here yielded similar errors to those from other studies for only clear sky conditions. This demonstrates potential for regional scale applications when reliable sources of solar radiation, air temperature and $T_{RAD}$ are available. A refined model for soil heat flux (G), based on the soil temperature-G relationship, was

evaluated from green-up to senescence using data for multiple years, and yielded errors half the magnitude of the standard TSEB formulation based on the relationship between $R_{NS}$ and G. The TSEB $\alpha_{PTC}$ parameterization for estimating canopy transpiration (LEc)  was tested using the standard TSEB value of 1.26 and a value of 0.92 suggested in the literature for Arctic tundra, and both parameterizations yield similar flux errors suggesting tundra-specific values of $\alpha_{PTC}$ are not needed. In the absence of in-situ measurements of LAI within the vicinity of the tower sites, the MODIS LAI product provided

reasonable inputs for localized model testing. The model was able to reproduce accurately temporal trends of energy partitioning in concert with tundra vegetation dynamics in the peak growing season. Moreover, it also has potential to monitor changes in surface energy fluxes in Arctic tundra due to changes in vegetation composition (e.g., shrub encroachment). This is particularly crucial in the Arctic where there is a sparse network of meteorological and flux observations.  Further research is needed regarding the specific role of the moss layer in modifying remote sensing estimates

of green vegetation cover fraction and soil heat conduction within tundra ecosystems. Future work will incorporate the TSEB model refinements identified here for Arctic tundra into regional and global applications of the ALEXI surface energy balance modelling system. Model performance within a fully satellite-based remote sensing framework will be compared to the local evaluations reported here at these tundra flux sites.  In addition, the diagnostic assessments of ET and surface energy fluxes will be compared with regional output from process-based

prognostic land-surface models to better understand the strengths and weaknesses of both types of modelling systems.

**Acknowledgements**

We would like to thank the anonymous reviewers for their insightful comments and suggestions, which we believe have significantly improved the quality and clarity of this manuscript. This research was supported by the Alaska NASA EPSCoR program awards NNX10NO2A and NNX13AB28A. Authors would also like to thank Colin Edgar from the Institute of Arctic Biology, UAF, for his help in data processing of the eddy covariance and meteorological data. Datasets from the Imnavait sites were provided by the Institute of Arctic Biology, UAF, based upon work supported by the National Science Foundation under grant #1107892. USDA is an equal opportunity employer and provider.

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

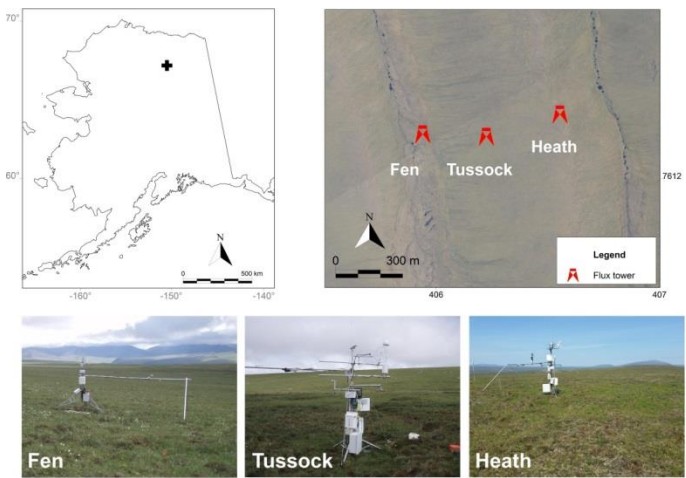

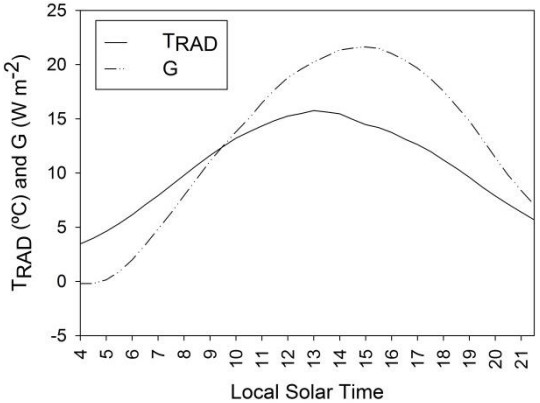

**Figure 1: Location of the Fen, Tussock and Heath flux towers at Imnavait watershed. Right panel map is in UTM-6N NAD83 with coordinates in km.**

5  **Figure 2. Mean daytime cycle for G and $T_{RAD}$ in the study area computed using all data available from the Fen, Tussock and Heath flux towers from 2008 to 2012.**

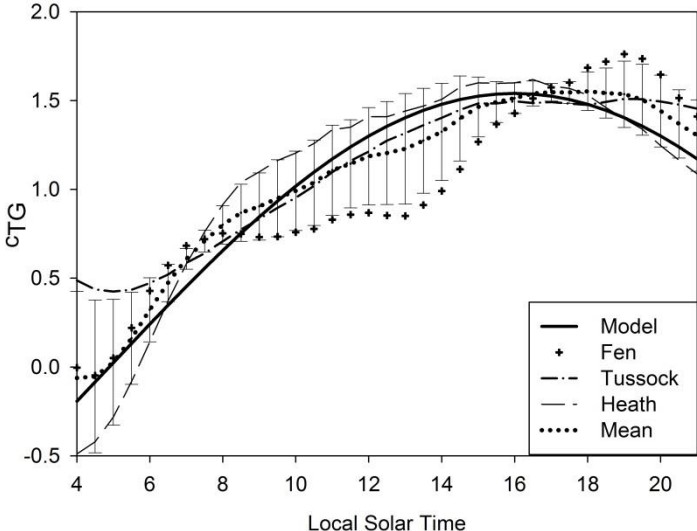

**Figure 3. Time series of modelled $c_{TG}$ and observed $c_{TG}$ values from the Fen, Tussock and Heath flux stations as well as mean values for summer conditions (bars represent standard deviation of the mean)**

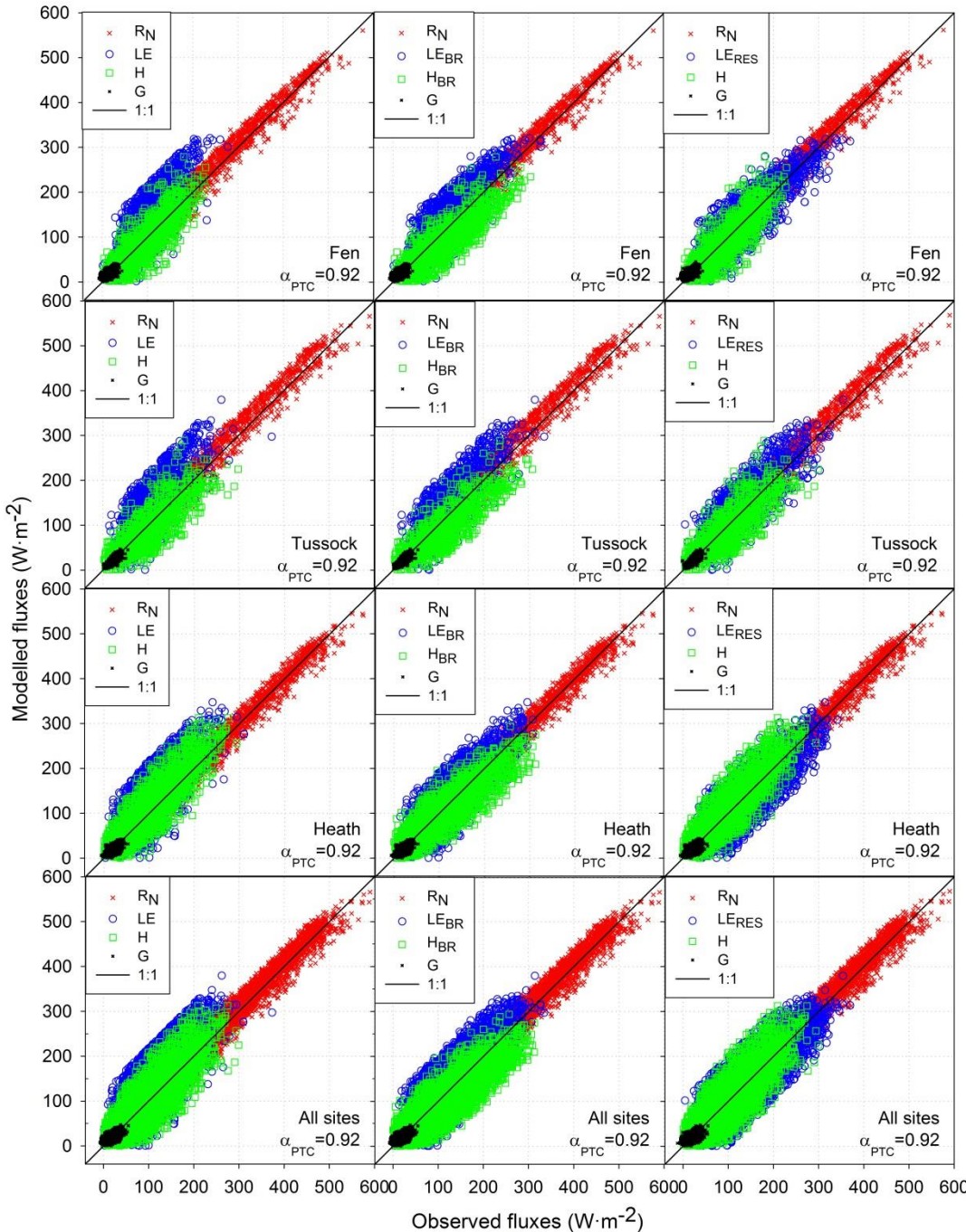

**Figure 4. Comparison of modelled vs. measured half-hourly fluxes using $\alpha_{PTC}$ of 0.92. The 1:1 line represents perfect agreement with observations. Columns represent results with unclosed observed turbulent fluxes (left), Bowen ratio closure (middle) and residual closure (right) for the fen, tussock, heath sites and all sites combined (rows).**

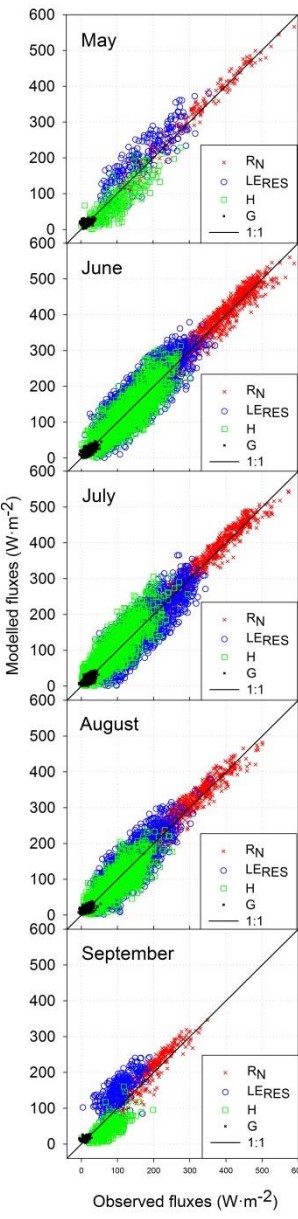

**Figure 5. Comparison of modelled vs. observed half-hourly surface fluxes (using LE from residual closure) by month using $\alpha_{PTC}$ of 0.92 and G estimated by the new model. The 1:1 line represents perfect agreement with observations.**

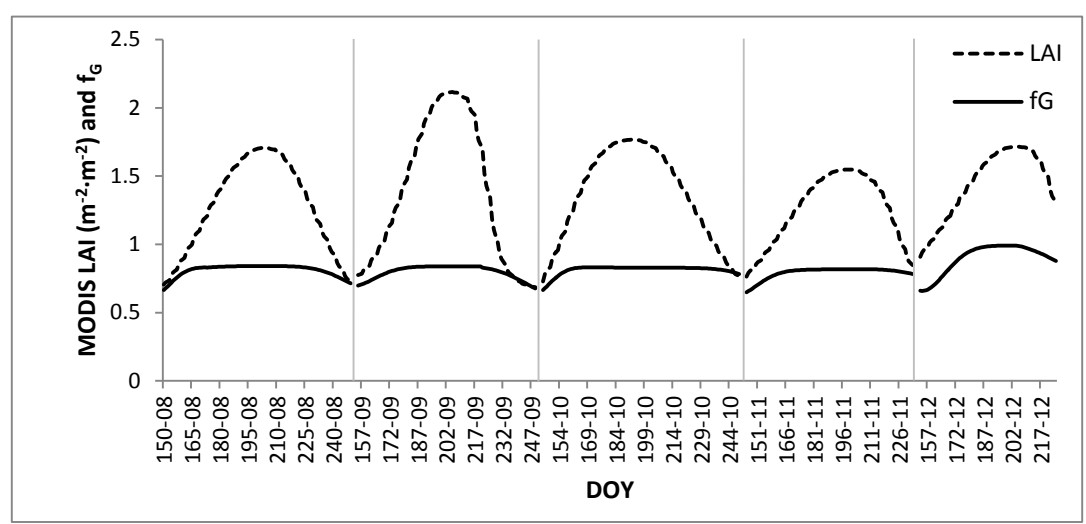

**Figure 6. Mean MODIS LAI and fraction of green vegetation (f$_G$) temporal dynamics for all flux stations from 2008 to 2012 .**

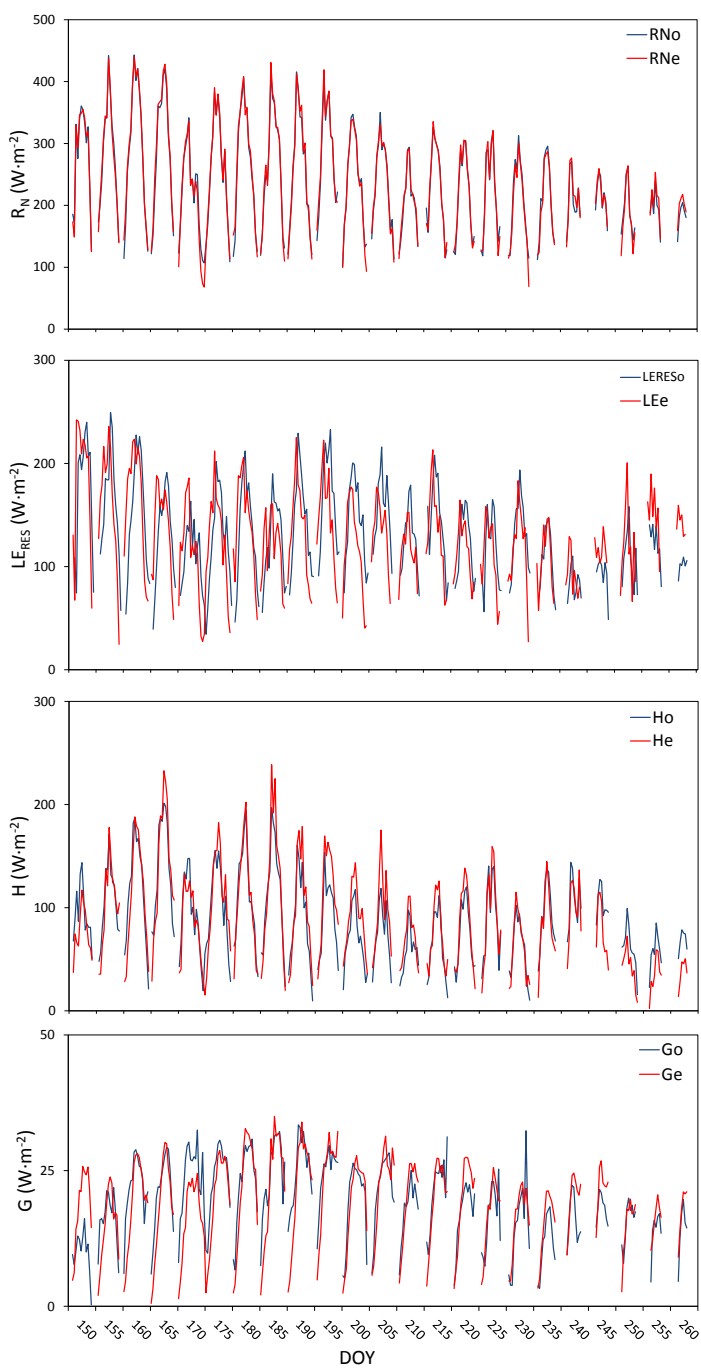

**Figure 7. Comparison of hourly flux tower RN, LE, H and G observations (using LE from residual closure) (o) (from 6 to 21 hours local solar time) at the Heath flux tower with model estimates (e) using αPTC of 0.92. Each diurnal segment represents flux data averaged by hour over 5-day intervals from 2008 to 2012.**

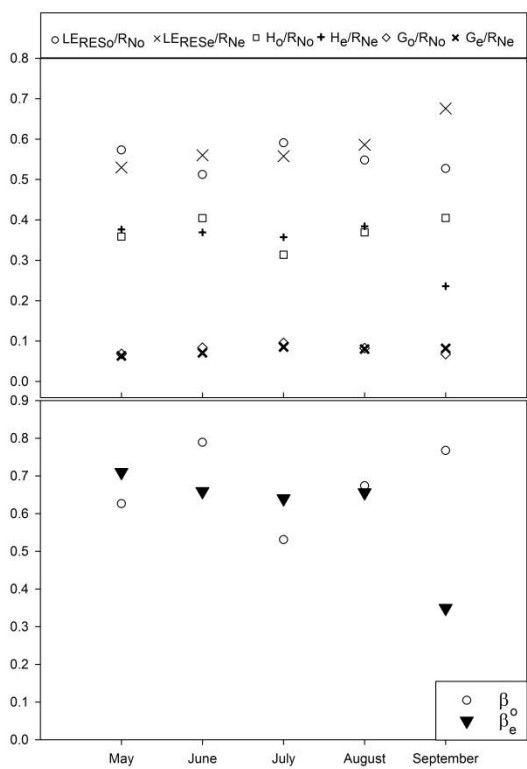

**Figure 8. Monthly mean observed (o) and estimated (e) energy partitioning (LE/RN, H/RN and G/RN) and Bowen ratio (β) for all flux stations from 2008 to 2012 using αPTC of 0.92.**

| Flux station name | | | Fen | Tussock | Heath |
|---|---|---|---|---|---|
| Coordinates (lat, long - WGS84) | | | 68.606, -149.311 | 68.606, -149.304 | 68.607, -149.296 |
| Period (Year|Day of Year) | | | 2008|194-252 | 2009|194-253 | 2008|194-252 |
| | | | 2010|142-262 | 2010|142-262 | 2009|159-253 |
| | | | 2011|217-262 | 2012|156-226 | 2010|143-262 |
| | | | 2012|153-264 | | 2011|147-262 |
| | | | | | 2012|156-226 |
| TSEB inputs | Symbol | Units | | | |
| Wind speed | $u$ | m s$^{-1}$ | 3.3 ± 1.7 | 3.4 ± 1.4 | 3.4 ± 1.4 |
| Air temperature | $T_a$ | °C | 11.6 ± 3.5 | 12.8 ± 3.8 | 12.8 ± 3.8 |
| Vapor pressure | $e_a$ | kPa | 0.9 ± 0.2 | 1 ± 0.3 | 1 ± 0.3 |
| Atmospheric pressure | $P$ | kPa | 92 ± 0.2 | 90 ± 0.6 | 90 ± 0.6 |
| Solar radiation | $S_d$ | W·m$^{-2}$ | 432 ± 121 | 503 ± 149 | 503 ± 149 |
| Longwave incoming radiation | $L_d$ | W·m$^{-2}$ | 261 ± 37 | 245 ± 34 | 245 ± 34 |
| Surface temperature | $T_s$ | K | 288 ± 4 | 290 ± 5 | 290 ± 6 |
| Leaf area index (MODIS) | $LAI$ | m$^2$·m$^{-2}$ | 1.15 ± 0.32 | 1.28 ± 0.42 | 1.25 ± 0.4 |
| Canopy height | $h_c$ | m | 0.4 | 0.4 | 0.4 |
| Clumping factor | $\Omega_c$ | | 1 | 1 | 1 |
| Fraction of green vegetation | $f_g$ | | 0.92 ± 0.01 | 0.9 ± 0.03 | 0.9 ± 0.03 |

**Table 1. Flux station name and location, period of model evaluation and list of inputs required by the TSEB. Average and standard deviation for the input values were computed for the full period of model evaluation for each site.**

| Instrument | Description | Height/*Depth*(m) |
|---|---|---|
| Campbell Sci. CSAT3 | Three Dimensional Sonic Anemometer | 2.18 - 3.18 |
| Licor LI-7500 | Open Path Infrared Gas Analyzer ($CO2$ and $H2O$) | 2.18 - 3.18 |
| Vaisalla HMP45C | Temperature and Relative Humidity Probe | 1.93 - 2.82 |
| Hukseflux HFP01SC | Self-calibrating Soil Heat Flux Plates (four per site) | *0.08* |
| Campbell Sci. TCAV | Type E Thermocouple Averaging Soil Temperature Probes (two per site) | *0.02-0.04* |
| Campbell Sci. CS616 | Water Content Reflectometers (two per site) | *0.025* |
| Licor LI190SB | PAR Sensor (incoming) | 2 - 3.6 |
| Licor LI190SB | PAR Sensor (outgoing) | 2 |

| Met One Ins. 014A | Wind Speed Sensor | 1.5 - 2.26 |
| Kipp & Zonen CMA6 | Pyranometer/Albedometer | 2 |
| *Kipp & Zonen CNR4 | Four components net Radiometer | 2 |
| Kipp & Zonen NR-Lite | Net radiation | 2 |
| Apogee IRR-P | InfraRed Radiometer Sensor | 1.5 - 3 |

**Table 2. General overview of the Fen, Tussock and Heath flux sites instrumentation (more information available at: http://aon.iab.uaf.edu/imnavait). Apogee infrared radiometers were oriented 45º off-nadir at the three flux stations. Asterisk (*) means that this instrument is only available at the Tussock flux station.**

| | | | SF03 | | | | | K98 | | | | |
|---|---|---|---|---|---|---|---|---|---|---|---|---|
| | $c_G$ | n | $R^2$ | RMSE | MBE | MAD | MADP | $R^2$ | RMSE | MBE | MAD | MADP |
| Fen | 0.30 | 1558 | 0.04 | 23 | 3 | 20 | 128 | 0.01 | 40 | 23 | 31 | 199 |
| | 0.14 | | 0.04 | 15 | -7 | 12 | 76 | 0.01 | 15 | 2 | 11 | 73 |
| Tussock | 0.30 | 1273 | 0.18 | 23 | 3 | 18 | 78 | 0.05 | 39 | 21 | 32 | 138 |
| | 0.14 | | 0.23 | 17 | -11 | 12 | 53 | 0.05 | 15 | -4 | 11 | 46 |
| Heath | 0.30 | 2347 | 0.11 | 26 | -5 | 20 | 96 | 0.10 | 34 | 14 | 26 | 125 |
| | 0.14 | | 0.14 | 21 | -14 | 15 | 72 | 0.06 | 16 | -5 | 10 | 48 |
| Total | 0.30 | 5178 | 0.12 | 25 | 0 | 20 | 98 | 0.03 | 37 | 19 | 29 | 145 |
| | 0.14 | | 0.10 | 18 | -11 | 14 | 68 | 0.03 | 15 | -3 | 11 | 53 |

**Table 3. Performance statistics for the soil heat flux estimation using Santanello and Friedl (2003), SF03, and Kustas et al. (1998), K98, methodologies and two values for the maximum $c_G$ value. RMSE, MBE and MAD are in W·m⁻² and MADP in %.**

| | Fit subset (60%) | | | | Test subset (40%) | | | | Flux dataset | | | |
|---|---|---|---|---|---|---|---|---|---|---|---|---|
| | Fen | Tussock | Heath | Total | Fen | Tussock | Heath | Total | Fen | Tussock | Heath | Total |
| $R^2$ | 0.89 | 0.99 | 0.99 | 0.99 | 0.55 | 0.77 | 0.69 | 0.68 | 0.27 | 0.56 | 0.49 | 0.44 |
| RMSE | 3.9 | 1 | 1 | 1 | 7 | 5 | 6 | 6 | 9 | 5 | 7 | 7 |
| MBE | 1.7 | -0.2 | -0.6 | 0.1 | 0.6 | -0.3 | -0.3 | 0 | 3.9 | 0.5 | -0.3 | 1 |
| MAD | 2.8 | 1 | 1 | 1 | 5 | 4 | 5 | 4 | 7 | 4 | 5 | 6 |
| MAPD | 25 | 8 | 8 | 8 | 49 | 28 | 38 | 37 | 44 | 17 | 24 | 28 |
| n | 8283 | 10332 | 10748 | 29363 | 3310 | 4122 | 4273 | 11705 | 1558 | 1273 | 2347 | 5178 |

**Table 4. Accuracy statistic for the new $c_{TG}$ approach for the fit and the test. RMSE, MBE and MAD in W·m⁻², MADP in % and n is number of half-hour intervals.**

| $R_N$ | n | $R^2$ | RMSE | MBE | MAD | MAPD |
|---|---|---|---|---|---|---|
| Fen | 1558 | 0.99 | 23 | 8 | 18 | 7 |
| Tussock | 1273 | 0.99 | 25 | 12 | 19 | 7 |
| Heath | 2347 | 0.99 | 20 | 2 | 15 | 6 |
| Total | 5178 | 0.99 | 23 | 7 | 18 | 7 |

| LE | n | $R^2$ | RMSE | MBE | MAD | MAPD |
|---|---|---|---|---|---|---|
| Fen | 1558 | 0.54 | 47 | 27 | 37 | 36 |
| Tussock | 1273 | 0.52 | 61 | 44 | 51 | 41 |
| Heath | 2347 | 0.55 | 54 | 33 | 44 | 42 |
| Total | 5178 | 0.54 | 53 | 35 | 44 | 40 |

| $LE_{BR}$ | n | $R^2$ | RMSE | MBE | MAD | MAPD |
|---|---|---|---|---|---|---|
| Fen | 1558 | 0.76 | 45 | 25 | 37 | 30 |
| Tussock | 1273 | 0.66 | 52 | 33 | 43 | 33 |
| Heath | 2347 | 0.65 | 43 | 15 | 35 | 28 |
| Total | 5178 | 0.71 | 46 | 23 | 38 | 35 |

| $LE_{RES}$ | n | $R^2$ | RMSE | MBE | MAD | MAPD |
|---|---|---|---|---|---|---|
| Fen | 1558 | 0.74 | 37 | 9 | 29 | 21 |
| Tussock | 1273 | 0.68 | 45 | 24 | 38 | 26 |
| Heath | 2347 | 0.68 | 39 | -3 | 31 | 21 |
| Total | 5178 | 0.69 | 40 | 7 | 32 | 30 |

| H | n | $R^2$ | RMSE | MBE | MAD | MAPD |
|---|---|---|---|---|---|---|
| Fen | 1558 | 0.66 | 28 | -6 | 22 | 23 |
| Tussock | 1273 | 0.65 | 33 | -12 | 26 | 24 |
| Heath | 2347 | 0.71 | 33 | 4 | 26 | 26 |
| Total | 5178 | 0.67 | 32 | -3 | 25 | 25 |

| $H_{BR}$ | n | $R^2$ | RMSE | MBE | MAD | MAPD |
|---|---|---|---|---|---|---|
| Fen | 1558 | 0.69 | 39 | -24 | 31 | 27 |
| Tussock | 1273 | 0.67 | 39 | -22 | 29 | 26 |
| Heath | 2347 | 0.72 | 38 | -14 | 30 | 25 |
| Total | 5178 | 0.69 | 39 | -19 | 31 | 30 |

**Table 5. Accuracy and error statistics from the comparison of modelled vs. observed unclosed and closed surface fluxes surface fluxes using $\alpha_{PTC}$ of 0.92. n is the number of half-hour periods analysed. RMSE, MAD and MBE are in $W \cdot m^{-2}$ and MADP in %.**

| | | | $R_N$ | | | | | | | | LE | | | |
|---|---|---|---|---|---|---|---|---|---|---|---|---|---|---|
| | n | $R^2$ | RMSE | MBE | MAD | MAPD | | n | $R^2$ | RMSE | MBE | MAD | MAPD |
| Fen | 1558 | 0.99 | 23 | 8 | 18 | 7 | Fen | 1558 | 0.66 | 58 | 45 | 52 | 45 |
| Tussock | 1273 | 0.99 | 25 | 12 | 19 | 7 | Tussock | 1273 | 0.54 | 59 | 52 | 50 | 48 |
| Heath | 2347 | 0.99 | 20 | 2 | 15 | 6 | Heath | 2347 | 0.52 | 57 | 39 | 47 | 42 |
| Total | 5178 | 0.99 | 23 | 7 | 18 | 7 | Total | 5178 | 0.59 | 58 | 45 | 50 | 45 |

| | | | $LE_{BR}$ | | | | | | | | $LE_{RES}$ | | | |
|---|---|---|---|---|---|---|---|---|---|---|---|---|---|---|
| | n | $R^2$ | RMSE | MBE | MAD | MAPD | | n | $R^2$ | RMSE | MBE | MAD | MAPD |
| Fen | 1558 | 0.73 | 52 | 36 | 43 | 35 | Fen | 1558 | 0.76 | 41 | 20 | 34 | 24 |
| Tussock | 1273 | 0.64 | 56 | 40 | 48 | 37 | Tussock | 1273 | 0.7 | 53 | 36 | 45 | 31 |
| Heath | 2347 | 0.66 | 48 | 26 | 40 | 32 | Heath | 2347 | 0.71 | 40 | 9 | 32 | 22 |
| Total | 5178 | 0.66 | 51 | 32 | 43 | 40 | Total | 5178 | 0.7 | 44 | 19 | 36 | 33 |

| | | | H | | | | | | | | $H_{BR}$ | | | |
|---|---|---|---|---|---|---|---|---|---|---|---|---|---|---|
| | n | $R^2$ | RMSE | MBE | MAD | MAPD | | n | $R^2$ | RMSE | MBE | MAD | MAPD |
| Fen | 1558 | 0.62 | 33 | -18 | 26 | 27 | Fen | 1558 | 0.64 | 46 | -34 | 38 | 33 |
| Tussock | 1273 | 0.6 | 39 | -24 | 31 | 29 | Tussock | 1273 | 0.62 | 43 | -29 | 35 | 28 |
| Heath | 2347 | 0.67 | 33 | -8 | 26 | 26 | Heath | 2347 | 0.65 | 43 | -25 | 35 | 28 |
| Total | 5178 | 0.64 | 35 | -15 | 28 | 27 | Total | 5178 | 0.64 | 44 | -29 | 36 | 36 |

**Table 6. Accuracy and error statistics from the comparison of modelled vs. observed unclosed and closed surface fluxes using $\alpha_{PTC}$ of 1.26. n is the number of half-hour periods analysed. RMSE, MAD and MBE are in W·m$^{-2}$ and MADP in %.**

| | | | $R_N$ | | | | LE | | | | |
|---|---|---|---|---|---|---|---|---|---|---|---|
| | n | $R^2$ | RMSE | MBE | MAD | MAPD | $R^2$ | RMSE | MBE | MAD | MAPD |
| May | 227 | 0.99 | 24 | 7 | 19 | 7 | 0.78 | 46 | 28 | 38 | 23 |
| June | 1727 | 0.99 | 22 | 6 | 17 | 6 | 0.73 | 40 | 11 | 32 | 20 |
| July | 1647 | 0.99 | 21 | 5 | 17 | 6 | 0.72 | 39 | -7 | 31 | 20 |
| August | 1264 | 0.99 | 23 | 6 | 19 | 8 | 0.64 | 37 | 7 | 30 | 24 |
| September | 312 | 0.99 | 26 | 14 | 23 | 12 | 0.44 | 52 | 39 | 46 | 45 |

| | | | H | | | | G | | | | |
|---|---|---|---|---|---|---|---|---|---|---|---|
| | n | $R^2$ | RMSE | MBE | MAD | MAPD | $R^2$ | RMSE | MBE | MAD | MAPD |
| May | 227 | 0.69 | 36 | -22 | 29 | 27 | 0.12 | 10 | 1 | 8 | 48 |
| June | 1727 | 0.71 | 32 | -6 | 25 | 20 | 0.45 | 7 | 0 | 6 | 26 |
| July | 1647 | 0.72 | 32 | 10 | 25 | 29 | 0.49 | 6 | 1 | 5 | 23 |
| August | 1264 | 0.62 | 37 | 7 | 30 | 24 | 0.40 | 7 | 3 | 6 | 34 |

| September | 312 | 0.39 | 38 | -31 | 32 | 42 | 0.27 | 7 | 4 | 5 | 40 |

**Table 7. Mean monthly accuracy and error statistics from the comparison of modelled vs. observed surface fluxes (using LE from residual closure) using αPTC of 0.92. n is the number of half-hour periods analysed. RMSE, MAD and MBE are in W·m⁻² and MADP in %.**

