# Peer review of "Estimation of surface energy fluxes in the Arctic tundra using the remote sensing thermal-based Two-Source Energy Balance model"

_Hydrology and Earth System Sciences, 2016_

## Referee Comment (RC1) · Anonymous Referee #1 · 20 Jul 2016

**General comments:**

The study analyses the performance of the Two-Source Energy Balance model at three flux tower locations at a Alaskan tundra site. The authors suggest and test model improvements concerning the parameterisations of incoming longwave radiation, soil heat flux, and latent heat flux. Overall they find that the most substantial improvement could be achieved with the adapted implementation of soil heat fluxes. Furthermore the model accuracy compared favourably with other models from literature and in different environments.

The study appears sound and the topic is very relevant as there is a lack of validated

surface energy balance models for Arctic tundra. The manuscript is well-written. However, I have questions about the methodology and I would be happy to see a more substantial discussion of some points. Furthermore, the title, abstract, motivation and conclusion emphasise satellite data, while the methodology only includes satellite LAI but no other satellite input. So either the methods are not complete, or the focus should be more local scale to avoid overselling the study. Nevertheless, I find the manuscript worth publishing in HESS after some revisions. Please do not get distracted by the number of points which I mention – I like the manuscript in general and I would like to stress that the work is really interesting. Good luck for the revisions!

**Major points:**

1. The authors stress the point that a remote-sensing based model can be applied at the larger scale (Title, Abstact p. 1, l. 17, 24, 26; Motivation p. 3, l. 1–12; Conclusions p. 15, l. 8–10). However, it seems that (except for the LAI, which is a minor point of the study) this was not done (p. 11, l. 23–24). This is a little bit disappointing after reading pages 1–3. Therefore I would suggest to force the model with satellite data only and compare the results. If this is beyond the scope of the paper, the authors should adjust the motivation statements.

2. Section 2 is quite long given that the model description is published already. P4 l20 – p5 l12 could be omitted or moved to an appendix as the resistance terms and the sensible heat flux parameterisation are not discussed further in the manuscript. In this case, you could mention after Equation 11 that $H_s$ is calculated as a function of the difference between canopy air temperature and soil temperature and of the soil resistance.

3. You show two different approaches for estimating $c_G$ (Section 3.2). In both approaches you fit some parameters. However, if I understand it correctly, you use

different data for fitting. On what data did you fit the parameters of the first method (p7, l23–24)? Why did you not use the same approach as for the second method, where you split the data set into a calibration and a validation subset? Are the data of all stations combined in a single data set? Do you take an equal amount of data points per station? Are the parameters fitted separately for month? Please describe the fitting approach in more detail in Section 3.2. Would it be possible to use a proxy such as soil moisture to improve the fit? Although you mention that soil type and properties are important, none of your methods takes it into account.

4. In Section 3.3 you describe that you use two different Priestley-Taylor coefficients. Did you consider varying them with soil moisture or LAI? Are they valid for the whole Arctic, or only locally?

5. Figure 2 does not demonstrate a relationship between $T_{RAD}$ and $G$, it merely shows that both variables exhibit a diel cycle (p11, l11–12 & p15, l1–2). Can you please provide more details on the expected relationship? I find that this is an important point as one of your main conclusions is that the approach using $T_{RAD}$ is better than using $R_N$. If I understand your reasoning correctly, you assume that the relationship between $T_{RAD}$ and $G$ holds for different vegetation types, times of the growing season and weather conditions. This point needs to be discussed in more detail. For example, a recent study by Juszak et al. (2016) showed that two different vegetation types with close to identical top soil temperatures differed in $G$ by a factor of 2. It would be great if you showed evidence for this relationship under different conditions. I would at least expect to see scatterplots of $T_{RAD}$ and $G$ as compared to $R_N$ and $G$ and correlation coefficients. Of course you can use shifted time series to account for the time lack.

6. The results and discussion in Section 6 are for all stations combined. However, it would be interesting to read about the different (or similar) accuracies at the

different vegetation types. This is particularly relevant if you want to conclude on vegetation dynamics and vegetation change (p14, l20–22). Figures 4 and 5 also reveal differences between the stations. For example $LE$ is strongly overestimated at the tussock site. Why?

7. Why do you discuss the accuracy of $R_N$ (p12, l8–16; p14, l6–11, l24, l30–31, most figures and tables) and not of the incoming longwave radiation alone? If you use the shortwave radiation budget and outgoing longwave radiation from measurements and just compute the incoming longwave radiation in your model, it would be surprising if you found a substantial difference in $R_N$. Did you use any of the remote sensing products (p12, l15–16) to justify your conclusion that 'this methodology scheme can be used to obtain reliable estimates of $R_N$'?

8. All Figure legends, scale bars and axis labels are far too small. Please increase the font size to about the same as the figure caption. Please also avoid to rotate the figures (in figures 6,8,9) and the axis labels.

**Specific comments:**

**p1, l19** What is unique about tundra conditions?

**p1, l24–25, Section 2** How did you test the usefulness of the MODIS LAI? Maybe it would be helpful to compare the results of the three towers concerning the different LAI. Also, did you test if the model is sensitive to LAI variations? Which fluxes are influenced by LAI in the model?

**p1, l29** Omitting 'Near-surface or shelter level' would make the starting sentence more catchy.

**p2, l2–4** Less references would be enough.

**p2, l15** Do you really mean 'inconsistent', or rather 'sparse'?

**p2, l18** What is an 'increase in peak vegetation'? Do you mean vegetation growth / activity / LAI?

**p2, l19** Do fires contribute to the greening? Maybe it would make sense to exchange the first two sentences of this paragraph.

**p2, l24–25** As shown in the recent paper by Williamson et al. (2016), the albedo effects of shrubs may not be as clear. Also, wet surfaces and sparsely vegetated water may have an even lower albedo than shrubs (Gamon et al., 2012).

**p4, l7** Does this mean that the model uses a spherical leaf angle distribution for all vegetation? How do the results change, if an erectophile distribution is used for the graminoid vegetation (fen, tussock tundra)?

**eq. 1, 4–12** It is a bit confusing that $R$ can be radiation or resistance, depending on the subscript. Maybe you could use 'r' for the resistance values?

**p5, l25** The abbreviation TIR is not explained. Additionally, this paragraph suggests that the satellite data is used for the study. If this is not the case, delete the clause 'when daytime TIR satellite imagery is typically acquired'.

**Section 3.1** Why do you continue using the Brutsaert (1975) formula? Two comparison studies on empirical parametrisations of incoming longwave radiation found that other formulars described the data better, namely the Dilley and O'Brien (1998) clear sky formula and the Unsworth and Monteith (1975) cloud correction (Flerchinger et al., 2009; Juszak and Pellicciotti, 2013).

**p7, l5 & p.7, l 25–29** Actually, in Eq. 12, not $R_N$ is used but $R_{NS}$. Please make more clear which variable you use. And if you adjusted the model in case you use $R_N$.

**p7, l8–14** Exchange this paragraph with the first paragraph.

**p7, l15–17** Split the sentence in two parts as the 'while' does not follow easily on the first part of the sentence.

**p7, l23–24** Why does this sentence not appear in the results section?

**p8, l13** Remove '1.2.1'.

**p9, l8** Are you sure you have several *Dryas* species (as indicated by spp)? Also, *Dryas* is a dwarf shrub species, so it would be more accurate to write '..., other dwarf shrubs, and lichen'.

**p9, l11** What do you mean by 'vegetation-based measurements'? Maybe replace the term with 'canopy structure' or 'vegetation properties'.

**p9, l21–22** Other comprehensive LAI data from close-by can be used as reference, e.g. Shaver and Chapin (1991); Shippert et al. (1995); Williams et al. (2001); Walker et al. (2003); Williams et al. (2006); Shaver et al. (2007); Sweet et al. (2015). In particular the study of Williams et al. (2006) has many details on different types. I am sure there are even more studies which measured LAI as the Imnavait Watershed and Toolik lake are very well studied.

**p9, l24–27** The method to estimate $f_G$ is not clear to me. How do you estimate the fraction of absorbed PAR by the green vegetation? Is it equal to $PAR_{incoming} - PAR_{reflected}$ in your model? This would also include PAR absorption by bare soil, dead plant material, mosses and other elements. Guzinski et al. (2013) actually suggests to use a different method, based on NDVI and EVI (as you mention on page 14). Do you have another reference that actually recommends the PAR ratio method?

**p9, l29** Can you explain your choice of 1 for the clumping factor in more detail? What is a 'variable organic layer'?

**p9, l30** Vegetation height and the clumping factor are not variable. Can you estimate the uncertainty you introduce with this assumption?

**p10, l1–2** The sentence about future work should be moved to the discussion or conclusions.

**p10, l12–13** Why do you restrict the modelling to daytime conditions? It would be interesting to also test if the model is able to reproduce values at night. I am aware, that the incoming longwave radiation depends on cloud cover. However, you could interpolate the cloud cover during the night. How did you assess the presence of precipitation?

**Section 5** Using five different error estimates does not add additional information as compared to using only three. In your results, you rarely mention MAD and the information of MAPD and RMSE is largely the same. It is not very intuitive that in your notation the mean of $e_i$ is $\bar{X}$. You could use $e_i$ and $\bar{E}$ or $x_i$ and $\bar{X}$ (and the corresponding notation for $o_i$ and $\bar{Y}$) instead.

**p11, l21 & Table 4** What is this flux subset? Please describe the choice of the subset in the methods.

**p11, l23–24** The first clause of the long sentence is out of place, it is an outlook and would fit better at the end of the conclusions.

**p12, l2** To which method do the $R^2$ and the RMSE value belong?

**p12, l1–7** You found that the new method was not better than the original Brutsaert (1975) formula. However, this does not necessarily imply that the Brutsaert (1975) method is good. I would like to see a discussion of limitations and other potential approaches.

**p12, l18** What is the 'evaluation subset'?

**p12, l30–32** The BR and RES methods need to be explained in the methods section. How does this description relate to the Priestley–Taylor approach you explain in the methods? Do the two methods refer to the canopy or the soil $LE$ (eq. 10, 11)?

**p13, l26** Is the fraction of vegetation cover not estimated from the PAR budget? Please explain this in the methods! How sensitive is the model to LAI?

**p13, l30** An LAI of 1.7 seems to be quite high for the Imnavait Watershed. Did you compare with other data such as (Shaver and Chapin, 1991; Shippert et al., 1995; Williams et al., 2001; Walker et al., 2003; Williams et al., 2006; Shaver et al., 2007; Sweet et al., 2015)? Which vegetation type had this extreme value?

**p13, l30** Is $f_G$ a sensitive parameter?

**p14, l26** As the interannual variability is not mentioned in the results, it should not be mentioned here.

**p15, l3** 'other models' is unclear. Do you mean '$G$ computation from $R_N$'?

**p15, l3** As some readers start with reading the conclusions, it would be good to repeat that $\alpha_{PTC}$ is used to estimate $ET$.

**p15, l6** Was the model sensitive to LAI? I would be surprised, as LAI (in the model) does not influence $ET$, albedo, or any of the other major fluxes. Otherwise this conclusion is not valid.

**p15, l8–10** On which result do you base this conclusion?

**p15, l11–14** This seems very abstract. Maybe you could rather conclude on how to integrate more satellite data to apply the model to the regional scale.

**Figure 2** The temperature is not in Kelvin. I do not think it makes sense to take the mean of all available data as the station with most data will contribute more and biases can occur, for example if the coldest station on average starts measuring later during the year. I would prefer one plot per station, or a completely different graph (as explained above).

**Figure 3** This graph is very important. However, it would be great if you could add uncertainties, or at least standard deviations.

**Figures 4–6** In the caption, PTC should be a subscript. This way of plotting does not allow an evaluation of $G$, one of your main focusses. Also, it is impossible to tell the accuracy of $LE$. I suggest to use just one variable per panel and indicate the point density with colour (heat map). As this will result in four times more panels, I suggest to remove Figure 5 as the additional information is small.

**Figure 7** The figure caption should be self explanatory. Please define $f_G$.

**Figure 8** I would prefer to see a sample time series to 5-day averages of multiple stations.

**Figure 9** Change the symbols to make the figure easier to read. With the tiny legend and the turned figure it is impossible. I would suggest to have the same symbol for the same variable, once filled (for observed) and once empty (for modelled).

**Table 1** Space missing between Longwave and incoming; the captions says 'Average and standard deviation for the input values were computed for each period and for each site.' However, there is just one value per site given. Which period is it for?

**Table 3** MAPD not MADP

**Table 5–6** One $H$ misses the subscript.

**References**

Brutsaert, W.: On a Derivable Formula for Long-Wave Radiation From Clear Skies, Water Resources Research, 11, 742–744, doi:10.1029/WR011i005p00742, 1975.

Dilley, A. C. and O'Brien, D. M.: Estimating downward clear sky long-wave irradiance at the surface from screen temperature and precipitable water, Quarterly Journal of the Royal Meteorological Society, 124, 1391–1401, doi:10.1002/qj.49712454903, 1998.

Flerchinger, G. N., Xaio, W., Marks, D., Sauer, T. J., and Yu, Q.: Comparison of algorithms for incoming atmospheric long-wave radiation, Water Resources Research, 45, W03 423, doi:10.1029/2008WR007394, 2009.

Gamon, J. A., Kershaw, G. P., Williamson, S., and Hik, D. S.: Microtopographic patterns in an arctic baydjarakh field: do fine-grain patterns enforce landscape stability?, Environmental Research Letters, 7, 015 502, doi:10.1088/1748-9326/7/1/015502, 2012.

Guzinski, R., Anderson, M. C., Kustas, W. P., Nieto, H., and Sandholt, I.: Using a thermal-based two source energy balance model with time-differencing to estimate surface energy fluxes with day-night MODIS observations, Hydrology and Earth System Sciences, 17, 2809–2825, doi:10.5194/hess-17-2809-2013, 2013.

Juszak, I. and Pellicciotti, F.: A comparison of parameterizations of incoming longwave radiation over melting glaciers: Model robustness and seasonal variability, Journal of Geophysical Research: Atmospheres, 118, 3066–3084, doi:10.1002/jgrd.50277, 2013.

Juszak, I., Eugster, W., Heijmans, M. M. P. D., and Schaepman-Strub, G.: Contrasting radiation and soil heat fluxes in Arctic shrub and wet sedge tundra, Biogeosciences, 13, 4049–4064, doi:10.5194/bg-13-4049-2016, 2016.

Shaver, G. R. and Chapin, III, F. S.: Production: Biomass Relationships and Element Cycling in Contrasting Arctic Vegetation Types, Ecological Monographs, 61, 1–31, http://www.jstor.org/stable/1942997, 1991.

Shaver, G. R., Street, L. E., Rastetter, E. B., van Wijk, M. T., and Williams, M.: Functional convergence in regulation of net $CO_2$ flux in heterogeneous tundra landscapes in Alaska and Sweden, Journal of Ecology, 95, 802–817, doi:10.1111/j.1365-2745.2007.01259.x, 2007.

Shippert, M. M., Walker, D. A., Auerbach, N. A., and Lewis, B. E.: Biomass and leaf-area index maps derived from SPOT images for Toolik Lake and Imnavait Creek areas, Alaska, Polar Record, 31, 147–154, doi:10.1017/S0032247400013644, 1995.

Sweet, S. K., Griffin, K. L., Steltzer, H., Gough, L., and Boelman, N. T.: Greater deciduous shrub

abundance extends tundra peak season and increases modeled net $CO_2$ uptake, Global Change Biology, 21, 2394–2409, doi:10.1111/gcb.12852, 2015.

Unsworth, M. H. and Monteith, J. L.: Long-wave radiation at the ground 1. Angular distribution of incoming radiation, Quarterly Journal of the Royal Meteorological Society, 101, 13–24, doi:10.1002/qj.49710142703, 1975.

Walker, D. A., Jia, G. J., Epstein, H. E., Raynolds, M. K., Chapin, III, F. S., Copass, C., Hinzman, L. D., Knudson, J. A., Maier, H. A., Michaelson, G. J., Nelson, F., Ping, C. L., Romanovsky, V. E., and Shiklomanov, N.: Vegetation–soil-thaw-depth relationships along a low-arctic bioclimate gradient, Alaska: synthesis of information from the ATLAS studies, Permafrost and Periglacial Processes, 14, 103–123, doi:10.1002/ppp.452, 2003.

Williams, M., Rastetter, E. B., Shaver, G. R., Hobbie, J. E., Carpino, E., and Kwiatkowski, B. L.: Primary production of an arctic watershed: An uncertainty analysis, Ecological Applications, 11, 1800–1816, doi:10.1890/1051-0761(2001)011[1800:PPOAAW]2.0.CO;2, 2001.

Williams, M., Street, L. E., van Wijk, M. T., and Shaver, G. R.: Identifying Differences in Carbon Exchange among Arctic Ecosystem Types, Ecosystems, 9, 288–304, doi:10.1007/s10021-005-0146-y, 2006.

Williamson, S. N., Barrio, I. C., Hik, D. S., and Gamon, J. A.: Phenology and species determine growing-season albedo increase at the altitudinal limit of shrub growth in the sub-Arctic, Global Change Biology, pp. n/a–n/a, doi:10.1111/gcb.13297, 2016.
* * *

---

## Referee Comment (RC2) · Anonymous Referee #2 · 15 Aug 2016

General:

There is no doubt that the northern high latitude regions are undergoing significant change and will bear the brunt of continued climate change. The authors attempt to evaluate a two source energy balance model using eddy covariance data over a range of vegetation types in the Arctic. They found that improvements in the net radiation, soil heat flux and canopy transpiration schemes were needed in this unique Arctic environment to improve model performance. This work is potentially significant as there are few tools available for exploring the impact of changes in surface fluxes due to vegetation or future environmental change at the site to regional scale.

The manuscript is very well written and in general the paper is robust but the authors

simply use an existing model and tweak a few parameters to get a better fit. I'm not convinced that there is much scientific value that this paper has added. In addition, there is much discussion about the error of the model and comparison to error rates from previous models. Much of the comparison with previous studies simply state that observed and estimated results are in line with previous studies in the Arctic tundra. What have we actually learned from this study? There is nothing in the discussion that highlights the significance or implications of the results. The discussion could benefit from exploring why the models did or did not do so well and what could be done to improve them. What are the processes that are important that are not being accurately simulated. How does the physical and ecological environment challenge the modelling? How does this sort of model add to our understanding? There is a good opportunity to enhance the study using model benchmarking. Also not sure how this would all be scaled up to the whole Arctic.

Although the subject is highly relevant, given the lack of insight from the discussion and other issues in the paper, I would recommend accept with major revision.

Specific: The authors articulate a good case for undertaking their research and there is adequate acknowledgement of the previous literature although a summary of previous Arctic modelling that is relevant to your choice of model would be advantageous. They then propose an aim to evaluate the performance of the model during the Arctic growing season. However, it is unclear to me as to why you are doing this and what the ultimate goal is? Could you articulate what the big picture implications are in the introduction? In addition, I think you need to add an argument as to why this particular model as there are so many potential models with different scales and different functions. Why not use a process-based land surface model where you can relate the differences in model versus obs with processes rather than in your case changing a few parameters to get a better fit?

The authors use measured shortwave radiation yet estimate long wave radiation from observed air and land surface temperatures. I would have thought that this is problem-

atic for Arctic environments and could result in a large error in the net radiation. Given that highly accurate net radiation and soil heat flux measurements are needed for this approach, what is error associated with estimating long wave radiation in the model? In addition, the authors assume that G is a constant fraction of net radiation. This assumption is untested and there is clearly a large uncertainty in the probable fraction into G due to differences in surface properties such as soil type and moisture conditions as the authors point out, but particularly also the composition and structure of the various organic layers which are ubiquitous across the Arctic. It is well understood that the properties of moss and organic materials in particular influence the thermal and hydrological properties of the soil greatly. Therefore, I would like to see a more formalised assessment of the relative uncertainty in the calculation of G and Rn.

The authors give a mean value of 0.14 for cG and 0.92 for alphaPTC over the Arctic tundra. There is a rather a lot of handwaving here to suggest a single value for the entire Arctic tundra. What was the range of values across different vegetation types in the Arctic tundra. What was the error around the mean for this value? In addition what is the influence of changing cover over the growing season on both these values?

Table 2 shows the TCAV at 2 cm but this is usually an integrated measure with probes at two and 4 cm. Please check this.

G is hard to measure. There is a great uncertainty in measurements of G in the tundra because traditional heat flux plates are made with an assumed thermal conductivity for loamy soils but we know in the tundra that this is primarily organic heat and moss which has a significantly lower thermal conductivity. Therefore self-calibrating heat flux plates or corrections are required. Can you quantify the uncertainty in your ground heat flux measurements which is an important term because it feeds directly into the energy balance?

The use of MODIS LAI is particularly problematic in Arctic areas and it has been noted that the largest discrepancies in MODIS LAI are at Arctic tundra sites where the MODIS

product overestimates woody cover proportions. Given that you have no LAI observations you cannot make any conclusions about how they relate to fPAR for example on page 13 line 30. What specific product was used, was it the 250 m resolution? What was the spatial extent of your footprint for this dataset and how does that relates to the spatial separation of your sites? Specifically which QC flags were used? How were gaps treated in the timeseries? Perhaps use MODIS fPAR. Given you have tower measurements of this you could validate the MODIS fPAR and assess the error here.

It is not clear as to how you distinguish between canopy and soil in these Arctic systems for the TSEB model. What do you define as soil and what is canopy? You have no significant woody vegetation to form a canopy in the first place. The surface layer consists of mosses, lichen, Forbes and shrubs and forms a continuous layer that cannot be partitioned into soil and canopy. I suspect in general you don't have any bare soil at your sites. Hence I'm not sure why you are using a two layer model here in the first place? Can you justify the use of a two layer model here? Therefore the assumption that fPAR is equivalent to fG is not robust. To use this you will need to demonstrate clearly that this is the case.

The description of the eddy covariance data is minimal. What software was used to process the data and what algorithms and parameters were used? Exactly what quality flags were filtered? Due to the importance of determining the energy balance components for this study it is crucial to provide a thorough analysis of energy balance closure at the different sites across different periods (i.e. daytime and daily). What percentage of data were excluded due to different quality control previously mentioned as well as the three criteria mentioned. How were gaps in the data filled and worthy gap filled data used in the analysis? The criteria of a surface energy balance closure of greater than 70% doesn't instill a lot of confidence in the measurements. I would assume from this that the energy balance closure is quite low. This is probably due to the difficulty in measuring the soil heat flux. It is well known that the belowground environment is complex including not just soil but also layers of peat and organic material as well as living

moss and lichen. Therefore your estimates of G will be highly underestimated and will result in a low energy balance closure. Discuss. How did you account for these in the correction of the soil heat flux plates? At what depth did you have the heat flux plates placed? I see they were 8 cm but is that below the surface in the moss? If so then your heat flux plates are not in soil but in organic material. You should use the appropriate bulk density not the soil bulk density. Also it appears that you only have one heat flux plate measurement per site which is insufficient given the spatial heterogeneity in the surface. As previously mentioned the thermal conductivity of the heat flux plate is manufactured to a standard soil which will not be representative of what you are measuring in. This will all result in very large errors in the observed soil heat flux. Please provide a thorough estimate of error and uncertainty for this particular important measurement.

Given the difficulty in measuring G and the errors associated with that it may be worth trying to take G as a residual of the surface energy balance.

In addition, what is the uncertainty (random and model) in the fluxes for each of the sites?

The measures of performance are relatively standard so I don't think you need to include the formulas here but just cite a previous reference.

The distribution of residual energy based on the Bowen ratio is not a common practice and the community in general prefers to see the original data being used. This is overwhelmingly important in this environment where there are very large errors in measurements of G and also Rn, both of which go into the available energy term. Errors in these will propagate into errors in the turbulent heat flux terms if you force them based on the bon ratio. Calculating LE as the residual of the surface energy balance equation is even more problematic as it is the sole term carrying all errors in the other terms. I would insist on redoing the analysis using only the original data and not presenting the other methods because they are so error prone.

As mentioned in the summary there is a lot of focus on model error and performance.

[Figure]

However, these comparisons are with often in different types of models in different ecosystems which is like comparing apples and oranges. Most published models will have some reasonable performance but we should move away from a simple reporting of the error to include better and more robust benchmarking of models. For example, this model could be compared against a simple empirical model to assess quantitatively whether the model performs any better than a simple model with local meteorological drivers. Recent papers have started to do and I suggest this is something that you could do to strengthen your paper. For example see:

Whitley, R., Beringer, J., Hutley, L., Abramowitz, G., De Kauwe, M. G., Duursma, R., Evans, B., Haverd, V., Li, L., Ryu, Y., Smith, B., Wang, Y.-P., Williams, M. and Yu, Q.: A model inter-comparison study to examine limiting factors in modelling Australian tropical savannas, Biogeosciences Discuss., 12(23), 18999–19041, doi:10.5194/bgd-12-18999-2015, 2015.

Luo, Y. Q., Randerson, J. T., Abramowitz, G., Bacour, C., Blyth, E., Carvalhais, N., Ciais, P., Dalmonech, D., Fisher, J. B., Fisher, R., Friedlingstein, P., Hibbard, K., Hoffman, F., Huntzinger, D., Jones, C. D., Koven, C., Lawrence, D., Li, D. J., Mahecha, M., Niu, S. L., Norby, R., Piao, S. L., Qi, X., Peylin, P., Prentice, I. C., Riley, W., Reichstein, M., Schwalm, C., Wang, Y. P., Xia, J. Y., Zaehle, S. and Zhou, X. H.: A framework for benchmarking land models, Biogeosciences, 9(10), 3857–3874, doi:10.5194/bg-9-3857-2012, 2012.

Page 14, line 3, the effect of what over the model? Mosses? In addition in this paragraph although you should not use the modus LA it is still consistent with seasonal growth of deciduous shrubs in particular. It is not inconsistent to have a constant fPAR where almost all incoming PAR is absorbed. The Arctic environment is highly adapted to absorbing as much energy as it can. As the leaf area of the shrubs increases during the summer the absorbed PAR is spread out amongst a greater leaf area but the fraction of fPAR remains the same.

[Figure]

Given this is a two layer model where are the results from the canopy and soil components. Do you even need a two layer model? Perhaps evaluate the usefulness of this type of model in this type of environment.

—————————————————

---

## Author Comment (AC1) · 6 Oct 2016

AC: We would like to thank the reviewer for their insightful comments and suggestions, which we believe have significantly improved the quality of this manuscript as well as our research.

RC_1: p9, l24–27 The method to estimate fG is not clear to me. How do you estimate the fraction of absorbed PAR by the green vegetation? Is it equal to PAR incoming - PARreflected in your model? This would also include PAR absorption by bare soil,dead plant material, mosses and other elements. Guzinski et al. (2013) actually suggests to use a different method, based on NDVI and EVI (as you mention on page 14). Do you have another reference that actually recommends the PAR ratio method?

[Figure]

AC_1: We would also like to note to the reviewer that we have accepted his/her suggestion to use the Guzinsky et al. (2013) method based on the EVI and NDVI to estimate the fG. According to Fisher et al. 2008, fG is defined by FAPAR / FIPAR where FAPAR is the fraction of PAR absorbed by green vegetation cover and FIPAR the fraction of PAR intercepted by total vegetation cover. Due to a lack of FAPAR observations, we estimated fG using only FIPAR as suggested by Anser (1998) and as the results show this might have caused an overestimation of fG at the beginning and at the end of the growing season contributing to model-measurement disagreement. Although this was not a major point according to the reviewer, we have re-run the model using Guzinski et al. (2013) approach to estimate fG. The new results yield a better model agreement, although it does not provide reliable fG values at the end of the season (mainly in September).

RC_2: 1. The authors stress the point that a remote-sensing based model can be applied at the larger scale (Title, Abstact p. 1, l. 17, 24, 26; Motivation p. 3, l. 1–12; Conclusions p. 15, l. 8–10). However, it seems that (except for the LAI, which is a minor point of the study) this was not done (p. 11, l. 23–24). This is a little bit disappointing after reading pages 1–3. Therefore I would suggest to force the model with satellite data only and compare the results. If this is beyond the scope of the paper, the authors should adjust the motivation statements.

This paper is focused on the local application with the tower micrometeorological and flux measurements representing local conditions in order to more reliably evaluate and refine the TSEB model for regional application to the Arctic tundra. As we stated in the conclusion section we will extend this research to regional scales using a TSEB-based model refined to be robust for the Arctic tundra using satellite inputs. To better clarify this objective, we have added text to the introduction motivating the need for localized testing in preparation for improvement of a regional satellite based energy balance model.

RC_2: 2. Section 2 is quite long given that the model description is published already.

P4 l20 – p5 l12 could be omitted or moved to an appendix as the resistance terms and the sensible heat flux parameterisation are not discussed further in the manuscript. In this case, you could mention after Equation 11 that Hs is calculated as a function of the difference between canopy air temperature and soil temperature and of the soil resistance.

AC_2: Given questions raised by the second reviewer about the model formulation, we decided to retain the discussion of the TSEB formulations needed to understand the resulting refinements required to obtain good results (see discussions in sections 3 and 6).

RC_3: 3.1. You show two different approaches for estimating cG (Section 3.2). In both approaches you fit some parameters. However, if I understand it correctly, you use different data for fitting. On what data did you fit the parameters of the first method (p7, l23–24)? Why did you not use the same approach as for the second method, where you split the data set into a calibration and a validation subset? Are the data of all stations combined in a single data set? Do you take an equal amount of data points per station? Are the parameters fitted separately for month? Please describe the fitting approach in more detail in Section 3.2.

AC_3: We have improved section 3.2, 4.2 and Tables 3 and 4 to clarify these points. The Kustas et al. (1998) and Santanello and Friedl (2003) methods were evaluated against the same dataset used to evaluate all fluxes that had restrictions for balance closure, among others (see section 4.2). To fit and test the new cTG approach, data from the previous dataset with no restriction of balance closure and from 4 to 21 hours local solar time was used. Coefficients A, B and S were derived using 60% of all available data aggregated in 30 min timesteps for the whole summer period and the remaining 40% of the data were reserved for model testing. Table 4 shows the amount of data points per station (n) to derive model coefficients and test the cTG approach. To calibrate the cTG, for the Tussock and Heath flux towers n is similar (around 10 000) while for the Fen tower, less data were available (n ∼8 000).

RC_4: 3.2. Would it be possible to use a proxy such as soil moisture to improve the fit?

AC_4: Soil heat flux plate measurements were corrected to account for soil heat storage using soil moisture from the water content reflectometers. This has been added to the text.

RC_5: 3.3. Although you mention that soil type and properties are important, none of your methods takes it into account. AC_5: We agree with the reviewer that soil type and properties are important to model G. In the original TSEB formulation, a simple approach based on the relationship between G and RNS was used (Eq. 13). This approach has less complexity and requires no soil texture and moisture information, which, unfortunately, is not routinely available over large areas. For continental-to-global applications of the TSEB, we are indeed finding that variations in the main parameters of the G formulation are required – for example over rock or desert sands. However, the modifications derived here help to better capture thermal characteristics of the tundra substrate.

RC_6: 4. In Section 3.3 you describe that you use two different Priestley-Taylor coefficients. Did you consider varying them with soil moisture or LAI? Are they valid for the whole Arctic, or only locally?

AC_6: The initial values of the Priestley-Taylor coefficients (PTC) we used in this paper were the originally proposed value of 1.26 for application of TSEB and a value of 0.92 averaged from the references found in the literature focused on Arctic tundra. As a starting point for the model we consider this range in PTC applicable for Arctic vegetation.

AC_6: In addition, to clarify how TSEB can adjust PTC for moisture conditions, the following paragraph has been added in section 2: "Under stress conditions, TSEB iteratively reduces ïĄąPTC from its initial value. The TSEB model requires both a solution to the radiative temperature partitioning (Eq. 2) and the energy balance (Eqs.

6 and 7), with physically plausible model solutions for soil and vegetation temperatures and fluxes. Non-physical solutions, such as daytime condensation at the soil surface (i.e., LES < 0), can be obtained under conditions of moisture deficiency. This happens because LEC is overestimated in these cases by the Priestley–Taylor parameterization, which describes potential transpiration. The higher LEC leads to a cooler TC and TS must be accordingly larger to satisfy Eq. (7). This drives HS high, and the residual LES from Eq. (11) goes negative. If this condition is encountered by the TSEB scheme, ïĄąPTC is iteratively reduced until LES $\sim$ 0 (expected for a dry soil surface). However there are instances where the vegetation is not transpiring at the potential rate but is not stressed due to its adaption to water and climate conditions (Agam et al., 2010) or the fact that not all the vegetation is green or actively transpiring (Guzinski et al., 2013)."

RC_7: 5. Figure 2 does not demonstrate a relationship between TRAD and G, it merely shows that both variables exhibit a diel cycle (p11, l11–12 & p15, l1–2). Can you please provide more details on the expected relationship? I find that this is an important point as one of your main conclusions is that the approach using TRAD is better than using RN. If I understand your reasoning correctly, you assume that the relationship between TRAD and G holds for different vegetation types, times of the growing season and weather conditions. This point needs to be discussed in more detail. For example, a recent study by Juszak et al. (2016) showed that two different vegetation types with close to identical top soil temperatures differed in G by a factor of 2. It would be great if you showed evidence for this relationship under different conditions. I would at least expect to see scatterplots of TRAD and G as compared to RN and G and correlation coefficients. Of course you can use shifted time series to account for the time lack. RC_7: Figure 2 The temperature is not in Kelvin. I do not think it makes sense to take the mean of all available data as the station with most data will contribute more and biases can occur, for example if the coldest station on average starts measuring later during the year. I would prefer one plot per station, or a completely different graph (as explained above).

AC_7: The axis title has been corrected. The relationship between TRAD and G and the definition of the new coefficient cGT has been explained in section 3.2 in which G is computed using Eq. 18. This method uses a phase shift proposed by Santanello and Friedl (2003) and is supported by the measurements illustrated in Figure 2. Figure 2 was only meant to show this phase shift and text in p11, l11–12 has been changed accordingly. Figure 3 and Table 4 show the behaviour of the new coefficient cGT derived from the TRAD-G relationship on a per station basis.

RC_7: 6. The results and discussion in Section 6 are for all stations combined. However, it would be interesting to read about the different (or similar) accuracies at the different vegetation types. This is particularly relevant if you want to conclude on vegetation dynamics and vegetation change (p14, l20–22). Figures 4 and 5 also reveal differences between the stations. For example LE is strongly overestimated at the tussock site. Why?

AC_7: Unfortunately, without detailed ground measurements to verify the assumed TSEB vegetation inputs (such as LAI), it is hard to identify any single factor that may have been a major cause for model-measurement disagreement, but overall the TSEB performance is considered satisfactory for all sites evaluated in this paper.

RC_8: 7. Why do you discuss the accuracy of RN (p12, l8–16; p14, l6–11, l24, l30–31, most figures and tables) and not of the incoming longwave radiation alone? If you use the shortwave radiation budget and outgoing longwave radiation from measurements and just compute the incoming longwave radiation in your model, it would be surprising if you found a substantial difference in RN. Did you use any of the remote sensing products (p12, l15–16) to justify your conclusion that 'this methodology scheme can be used to obtain reliable estimates of RN'?

AC_8: Downwelling longwave radiation results were discussed at the beginning of section 6.2 before discussing RN results. AC_8: We have not used remote sensing products to justify this conclusion. This sentence has been rewritten accordingly.
RC_9: 8. All Figure legends, scale bars and axis labels are far too small. Please increase the font size to about the same as the figure caption. Please also avoid to rotate the figures (in figures 6,8,9) and the axis labels.

AC_9: Figure legends, scale bars and axis labels have been increased. AC_9: Figures 6, 8 and 9 have been re-rotated.

RC_10: p1, l19 What is unique about tundra conditions?

AC_10: "Unique" was misplaced. It was supposed to be written before "parameterizations". In any case it has been removed from the text to avoid leading to misinterpretations.

RC_11: p1, l24–25, Section 2 How did you test the usefulness of the MODIS LAI? Maybe it would be helpful to compare the results of the three towers concerning the different LAI. Also, did you test if the model is sensitive to LAI variations? Which fluxes are influenced by LAI in the model?

AC_11: Unfortunately, we do not have LAI field measurement, thus, MODIS LAI usefulness was tested indirectly by means of the evaluation of the surface energy fluxes. The model is sensitive to LAI, since the radiation and temperature partitioning are affected by the LAI/fractional cover as well as the wind speed at the soil surface and LEc via the PT parameterization for the Rnc (Timmermans et al., 2007).

RC_12: p9, l21–22 Other comprehensive LAI data from close-by can be used as reference, e.g. Shaver and Chapin (1991); Shippert et al. (1995); Williams et al. (2001); Walker et al. (2003); Williams et al. (2006); Shaver et al. (2007); Sweet et al. (2015). In particular the study of Williams et al. (2006) has many details on different types. I am sure there are even more studies which measured LAI as the Imnavait Watershed and Toolik lake are very well studied. RC_12: p13, l30 An LAI of 1.7 seems to be quite high for the Imnavait Watershed. Did you compare with other data such as (Shaver and Chapin, 1991; Shippert et al., 1995; Williams et al., 2001; Walker et al., 2003; Williams
et al., 2006; Shaver et al., 2007; Sweet et al., 2015)? Which vegetation type had this extreme value?

AC_12: References about reported LAI values in these previous works and alternative methods to estimate LAI in the Arctic tundra have been added.

RC_13: p1, l29 Omitting 'Near-surface or shelter level' would make the starting sentence more catchy.

AC_13: This has been deleted from the text.

RC_14: p2, l2–4 Less references would be enough.

AC_14: We have kept more recent and relevant publications.

RC_15: p2, l15 Do you really mean 'inconsistent', or rather 'sparse'?

AC_15: We meant spatially and temporally inconsistent, and we did not imply that the data is wrong in any sense. We have rewritten the sentence to avoid misinterpretations.

RC_16: p2, l18 What is an 'increase in peak vegetation'? Do you mean vegetation growth / activity / LAI?

AC_16: According to Jia et al., 2003 it is in the "peak vegetation greenness". The reference was misplaced. This has been changed in the text.

RC_17: p2, l19 Do fires contribute to the greening? Maybe it would make sense to exchange the first two sentences of this paragraph.

AC_17: Sentences have been exchanged in the text.

RC_18: p2, l24–25 As shown in the recent paper by Williamson et al. (2016), the albedo effects of shrubs may not be as clear. Also, wet surfaces and sparsely vegetated water may have an even lower albedo than shrubs (Gamon et al., 2012).

AC_18: We agree with the reviewer that wet surfaces and sparsely vegetated water may have an even lower albedo than shrubs. A reference to the Williamson et al.

[Figure]

paper has been added to the paper.

RC_19: p4, l7 Does this mean that the model uses a spherical leaf angle distribution for all vegetation? How do the results change, if an erectophile distribution is used for the graminoid vegetation (fen, tussock tundra)?

AC_19: The assumed leaf angle distribution will affect the radiation divergence through the canopy layer and hence affect the net radiation partitioning between the canopy overstory and the soil/substrate. Without measurements to determine the leaf angle distribution, the default of a spherical leaf angle distribution is a reasonable one, particularly for heterogeneous surfaces having a mixture of vegetation species.

RC_20: eq. 1, 4–12 It is a bit confusing that R can be radiation or resistance, depending on the subscript. Maybe you could use 'r' for the resistance values?

AC_20: "r" has been adopted for resistance and changed in the text.

RC_21: p5, l25 The abbreviation TIR is not explained. Additionally, this paragraph suggests that the satellite data is used for the study. If this is not the case, delete the clause 'when daytime TIR satellite imagery is typically acquired'.

AC_21: we have expanded the TIR abbreviation. This paragraph refers to the original method development in which cG=0.3 was set. However, in order to avoid misinterpretations we have deleted "when daytime thermal satellite imagery is typically acquired" from the paragraph.

RC_22: Section 3.1 Why do you continue using the Brutsaert (1975) formula? Two comparison studies on empirical parametrisations of incoming longwave radiation found that other formulars described the data better, namely the Dilley and O'Brien (1998) clear sky formula and the Unsworth and Monteith (1975) cloud correction (Flerchinger et al., 2009; Juszak and Pellicciotti, 2013).

AC_22: Although there are other sky emissivity parameterizations which might give slightly better estimates of incoming longwave, the error in using Brustaert formulation

in TSEB is minor compared to the errors in turbulent flux estimation. In fact from Table 5 in Flerchinger et al (2009) the RMSD from all sites measuring incoming longwave using Brutsaert (1975) is 27.2 W/m2 while for Dilley and O'Brien (1998) it is 23.3 W/m2. Regarding cloud correction, the Crawford and Duchon method is easier to apply since we do not have the data required for Unsworth and Monteith (1975) method.

RC_23: p7, l5 & p.7, l 25–29 Actually, in Eq. 12, not RN is used but RNS. Please make more clear which variable you use. And if you adjusted the model in case you use RN.

AC_23: This has been corrected in the text.

RC_24: p7, l8–14 Exchange this paragraph with the first paragraph.

AC_24: This has been exchanged in the text.

RC_25: p7, l15–17 Split the sentence in two parts as the 'while' does not follow easily on the first part of the sentence.

AC_25: This has been corrected in the text.

RC_26: p7, l23–24 Why does this sentence not appear in the results section?

AC_26: These are the values from the original model. We have rewritten the sentence to clarify the text.

RC_27: p8, l13 Remove '1.2.1'.

AC_27: This has been removed from the text.

RC_28: p9, l8 Are you sure you have several Dryas species (as indicated by spp)? Also, Dryas is a dwarf shrub species, so it would be more accurate to write '..., other dwarf shrubs, and lichen'.

AC_28: This has been modified in the text accordingly.

RC_29: p9, l11 What do you mean by 'vegetation-based measurements'? Maybe replace the term with 'canopy structure' or 'vegetation properties'.

AC_29: We have changed the section title using "vegetation properties".

RC_30: p9, l29 Can you explain your choice of 1 for the clumping factor in more detail? What is a 'variable organic layer'?

AC_30: It is not an organic layer, it is a moss layer, and this has been changed in the text. As text says clumping factor was set to 1 based on the knowledge that Arctic tundra has a variable moss layer with little bare ground, thus, almost covering almost 100% of the ground. We used this approach for modelling purposes as we do not have actual data on the ground. However, a value of 1 seems a realistic approach for the study area.

RC_31: p9, l30 Vegetation height and the clumping factor are not variable. Can you estimate the uncertainty you introduce with this assumption?

AC_31: Over the growing season ground measurements indicated little change in vegetation height and density. Prior sensitivity studies (e.g., Zhan et al., 1996) indicate TSEB shows relatively small sensitivity to canopy height and fractional cover, which is related to the vegetation clumping factor.

RC_32: p10, l1–2 The sentence about future work should be moved to the discussion or conclusions.

AC_32: This sentence has been moved to the conclusions section.

RC_33: p10, l12–13 Why do you restrict the modelling to daytime conditions? It would be interesting to also test if the model is able to reproduce values at night. I am aware, that the incoming longwave radiation depends on cloud cover. However, you could interpolate the cloud cover during the night. How did you assess the presence of precipitation?

AC_33: Our testing is focused on daytime conditions for two reasons: First, EC flux observations used for validation are less reliable during night-time due to stable conditions and low wind speeds. Second, for transition to satellite applications, we are

primarily interested in evaluating model performance during daytime satellite overpass times. Other techniques are typically used to upscale from the overpass time to daily total fluxes.

RC_34: Section 5 Using five different error estimates does not add additional information as compared to using only three. In your results, you rarely mention MAD and the information of MAPD and RMSE is largely the same. It is not very intuitive that in your notation the mean of ei is _X . You could use ei and _E or xi and _X (and the corresponding notation for oi and _ Y ) instead.

AC_34: We have used five different error estimates to make the results section more comparable to other papers. Although we agree with the reviewer, in the literature you may find some studies in which MAE or MAD are only stated.

AC_34: ei and oi notations have been changed in the text.

RC_35: p11, l21 & Table 4 What is this flux subset? Please describe the choice of the subset in the methods.

AC_35: This was clarified in section 4.2 "Model inputs and evaluation dataset" and Tables 3 and 4.

RC_36: p11, l23–24 The first clause of the long sentence is out of place, it is an outlook and would fit better at the end of the conclusions.

AC_36: Sentence has been move to the conclusions section.

RC_37: p12, l2 To which method do the R2 and the RMSE value belong?

AC_37: Both methods yielded similar results. R2 was the same and RMSE for Brustaert (1975) and Jin et al. (2006) was 26 Wm-2 and 27 Wm-2, respectively. This has been clarified in the text.

RC_38: p12, l1–7 You found that the new method was not better than the original Brutsaert (1975) formula. However, this does not necessarily imply that the Brutsaert

(1975) method is good. I would like to see a discussion of limitations and other potential approaches.

AC_38: Differences between methods for estimating clear sky incoming longwave radiation continue to be evaluated over different climate zones (e.g., Choi et al., 2008) and indicate that discrepancies tend to be relatively small compared to uncertainty in modelling the turbulent fluxes. Therefore, a detailed discussion is not warranted for this analysis (see also response above). RC_39: p12, l18 What is the 'evaluation subset'?

AC_39: This has been clarified in section 4.2 "Model inputs and evaluation datasets".

RC_40: p12, l30–32 The BR and RES methods need to be explained in the methods section. How does this description relate to the Priestley–Taylor approach you explain in the methods? Do the two methods refer to the canopy or the soil LE (eq. 10, 11)?

AC_40: BR (Bowen Ratio) and RES (Residual) methods have been referenced in the previous paragraph and they are intended to address the lack of closure of the flux station data used to evaluate the TSEB method. We compare TSEB to closed fluxes since the model requires energy balance closure while the measurements of H and LE using eddy covariance technique undermeasure these fluxes by 10-20% based on comparison with available energy (Rn-G). We used two methods: 1-a distribution of residual according to Bowen Ratio, with the acronym BR (Twine et al. 2000 and Foken 2008); 2- and LE was recalculated as the residual, with the acronym RES (Li et al., 2008). In order to clarify the text for these methods, we have introduces these acronyms in the previous sentence. These methods are well explained in these papers and, for the sake of brevity, we prefer to refer the reader to the original references.

RC_41: p13, l26 Is the fraction of vegetation cover not estimated from the PAR budget? Please explain this in the methods! How sensitive is the model to LAI?

AC_41: The fraction of vegetation cover (Eq. 3) is computed using LAI and not PAR. We have clarified this in the text.

RC_42: p13, l30 Is fG a sensitive parameter?

AC_42: The value of fG modifies the estimated canopy transpiration (LEC) via the Priestley-Taylor parameterization (Eq. 10). It reduces LEC in direct proportion to its magnitude and has been used to adjust LEC based on crop phenology in other studies (e.g., Guzinski et al., 2015).

RC_43: p14, l26 As the interannual variability is not mentioned in the results, it should not be mentioned here.

AC_43: We have replaced "interannual" by "seasonal"

RC_44: p15, l3 'other models' is unclear. Do you mean 'G computation from RN'?

AC_44: Yes, this has been clarified in the text.

RC_45: p15, l3 As some readers start with reading the conclusions, it would be good to repeat that _PTC is used to estimate ET.

AC_45: This has been added in the text.

RC_46: p15, l6 Was the model sensitive to LAI? I would be surprised, as LAI (in the model) does not influence ET, albedo, or any of the other major fluxes. Otherwise this conclusion is not valid.

AC_46: LAI is used by TSEB (Eq. 3) to partition TRAD into soil and canopy temperature components, thus, it influences surface energy flux partitioning between the canopy and soil/substrate. The value of LAI also influences the radiation divergence and wind profile through the canopy layer and ultimately the soil and canopy aerodynamic resistances (Kustas and Norman, 1999;2000).

RC_47: p15, l8–10 On which result do you base this conclusion?

AC_47: We base this conclusion on the fact that the remote sensing-based TSEB model is able to capture the vegetation seasonal dynamics and contains the main
factors (LST, LAI, vegetation height/roughness) affecting H and LE partitioning. Thus with a multi-year time series of remote sensing observations from satellites are able to detect changes in vegetation cover conditions (LAI, canopy height and roughness) which in turn can affect LST and hence energy flux partitioning. This permits monitoring the impact of vegetation cover changes on the water and energy cycle at synoptic scales with satellite data.

RC_48: p15, l11–14 This seems very abstract. Maybe you could rather conclude on how to integrate more satellite data to apply the model to the regional scale.

AC_48: Methods described in this sentence are designed to estimate surface energy fluxes with satellite data. We have clarified this in the text.

RC_49: Figure 3 This graph is very important. However, it would be great if you could add uncertainties, or at least standard deviations.

AC_49: Standard deviations for the mean values have been added to this figure.

RC_50: Figures 4–6 In the caption, PTC should be a subscript. This way of plotting does not allow an evaluation of G, one of your main focusses. Also, it is impossible to tell the accuracy of LE. I suggest to use just one variable per panel and indicate the point density with colour (heat map). As this will result in four times more panels, I suggest to remove Figure 5 as the additional information is small.

AC_50: PTC has been subscripted. AC_50: Difference statistics between modelled and measured energy balance components are provided in tables 3 through 6. Having separate graphs comparing LE, H, RN and G would make it more difficult for the reader to have a sense of the relative magnitudes and scatter between the measured and modelled energy balance components. Showing the results in this manner gives the reader a better sense of the relative modelled-measured differences and which fluxes is the scatter the largest and most significant in the four components.

RC_51: Figure 7 The figure caption should be self explanatory. Please define fG.

AC_51: This has been added in the caption.

RC_52: Figure 8 I would prefer to see a sample time series to 5-day averages of multiple stations.

AC_52: 5-day averaged fluxes displayed in the figures more readily indicates the seasonal behaviour of TSEB over the whole study period. A sample time series is too noisy and does not allow the seasonal dynamics of surface energy fluxes and energy partitioning to be easily determined or illustrated.

RC_53: Figure 9 Change the symbols to make the figure easier to read. With the tiny legend and the turned figure it is impossible. I would suggest to have the same symbol for the same variable, once filled (for observed) and once empty (for modelled).

AC_53: The figure has been turned, the legend has been increased in size and the symbols have been refilled.

RC_54: Table 1 Space missing between Longwave and incoming; the captions says 'Average and standard deviation for the input values were computed for each period and for each site.' However, there is just one value per site given. Which period is it for?

AC_54: This has been corrected in the text. AC_54: Average and standard deviations reported in this table were computed using all selected data from the full period of model evaluation (Period row) for each flux station. This has been clarified in the caption.

RC_55: Table 3 MAPD not MADP

AC_55: This has been corrected in the text.

RC_56: Table 5–6 One H misses the subscript.

AC_56: Sensible heat (H) is not missing the subscript. When using the residual method observed (from the flux tower) H is evaluated against modelled H.

References Asner, G. P.: Biophysical and Biochemical Sources of Variability in Canopy Reflectance, Remote Sens Environ, 64, 234–253, 1998.

Choi, M. H., Jacobs, J. M., and Kustas, W. P.: Assessment of clear and cloudy sky parameterizations for daily downwelling longwave radiation over different land surfaces in Florida, USA, Geophys Res Lett, 35, Artn L20402 10.1029/2008gl035731, 2008.

Fisher, J. B., Tu, K. P., and Baldocchi, D. D.: Global estimates of the land-atmosphere water flux based on monthly AVHRR and ISLSCP-II data, validated at 16 FLUXNET sites, Remote Sens Environ, 112, 901-919, 10.1016/j.rse.2007.06.025, 2008. Guzinski, R., Nieto, H., Stisen, S., and Fensholt, R.: Inter-comparison of energy balance and hydrological models for land surface energy flux estimation over a whole river catchment, Hydrol Earth Syst Sc, 19, 2017-2036, 10.5194/hess-19-2017-2015, 2015.

Kustas, W. P. and Norman, J. M. Evaluation of soil and vegetation heat flux predictions using a simple two-source model with radiometric temperatures for partial canopy cover. Agricultural and Forest Meteorology. 94:13-29. 1999.

Kustas, W. P. and Norman, J. M. A two-source energy balance approach using directional radiometric temperature observations for sparse canopy covered surfaces. Agronomy Journal. 92:847-854. 2000.

Timmermans, W. J., Kustas, W. P., Anderson, M. C., and French, A. N.: An inter-comparison of the Surface Energy Balance Algorithm for Land (SEBAL) and the Two-Source Energy Balance (TSEB) modeling schemes, Remote Sens Environ, 108, 369-384, 10.1016/j.rse.2006.11.028, 2007.

Zhan, X., Kustas, W. P., and Humes, K. S.: An intercomparison study on models of sensible heat flux over partial canopy surfaces with remotely sensed surface temperature, Remote Sens Environ, 58, 242-256, Doi 10.1016/S0034-4257(96)00049-1, 1996.

---

## Author Comment (AC2) · 6 Oct 2016

AC: We would like to thank the reviewer for their insightful comments and suggestions, which we believe have significantly improved the quality of this manuscript as well as our research. We would also like to note to the reviewer that we have accepted the suggestion of the other reviewer to use the Guzinsky et al. (2013) method based on the EVI and NDVI to estimate the fG. According to Fisher et al. 2008, fG is defined by FAPAR / FIPAR where FAPAR is the fraction of PAR absorbed by green vegetation cover and FIPAR the fraction of PAR intercepted by total vegetation cover. Due to a lack of FAPAR observations, we estimated fG using only FIPAR as suggested by Anser (1998) and as the results show this might have caused an overestimation of fG at the

beginning and at the end of the growing season contributing to model-measurement disagreement. Although this was not a major point according to the reviewer, we have re-run the model using Guzinsky et al. (2013) approach to estimate fG. The new results yielded better model agreement, although it does not provide reliable fG values at the end of the season (mainly in September) and further research needs to address this issue.

RC_1: The authors articulate a good case for undertaking their research and there is adequate acknowledgement of the previous literature although a summary of previous Arctic modelling that is relevant to your choice of model would be advantageous. They then propose an aim to evaluate the performance of the model during the Arctic growing season. However, it is unclear to me as to why you are doing this and what the ultimate goal is? Could you articulate what the big picture implications are in the introduction? In addition, I think you need to add an argument as to why this particular model as there are so many potential models with different scales and different functions. Why not use a process-based land surface model where you can relate the differences in model versus obs with processes rather than in your case changing a few parameters to get a better fit?

AC_1: We agree with the reviewer that the big picture motivation for evaluating TSEB performance over the Arctic tundra was not well described in the original submission, nor was our vision for upscaling to regional coverage. Our motivation is now better described in the final paragraphs of the introduction. In short, the TSEB forms the land surface model in a regional remote sensing energy balance system (ALEXI), used to model energy fluxes and ET from continental to global scales. ALEXI is currently used in NOAA OSPO's GET-D modeling system for North America (http://www.ospo.noaa.gov/Products/land/getd/index.html), and a prototype global modeling system is under development. ALEXI output has been evaluated over CONUS and lower latitude sites in Europe, but has not to date been tested over tundra ecosystems – constituting a significant fraction of the global land cover. Our primary goal in this paper is to evaluate TSEB performance over tundra, and to identify refinements that could be incorporated into the regional/global ALEXI system. This motivation is now more clearly outlined in the introduction. We also provide a rationale for investigating a diagnostic flux system, which can be compared in future studies to process-based prognostic model output. Hain et al. (2015) performed a comparison of ALEXI and Noah latent heat flux estimates over CONUS and found the TSEB was able to diagnose missing moisture source/sink processes in the prognostic model (e.g., due to irrigation, shallow groundwater, etc). This motivation for focus on a diagnostic approach is also now provided within the introduction.

AC_1: TSEB has been already compared with other methodologies showing superior performance (e.g., Timmermans et al., 2007; Choi et al., 2009; Tang et al., 2011). This has also been included in the text.

RC_2: The authors use measured shortwave radiation yet estimate long wave radiation from observed air and land surface temperatures. I would have thought that this is problematic for Arctic environments and could result in a large error in the net radiation. Given that highly accurate net radiation and soil heat flux measurements are needed for this approach, what is error associated with estimating long wave radiation in the model?

AC_2: Upwelling longwave radiation was computed using TRAD from the four component net radiation sensor and the Apogee IR sensors in each tower. Downwelling longwave radiation computed through Eq. 13 and estimation errors were reported in section 6.2, and showed a RMSE of 26 W.m-2.

RC_3: In addition, the authors assume that G is a constant fraction of net radiation. This assumption is untested and there is clearly a large uncertainty in the probable fraction into G due to differences in surface properties such as soil type and moisture conditions as the authors point out, but particularly also the composition and structure of the various organic layers which are ubiquitous across the Arctic. It is well understood that the properties of moss and organic materials in particular influence the thermal and hydrological properties of the soil greatly. Therefore, I would like to see a more formalised assessment of the relative uncertainty in the calculation of G and Rn.

AC_3: In the original TSEB formulation, a simple approach based on the relationship between G and RNS was used (Eq. 13). For continental-to-global applications of the TSEB, we are indeed finding that variations in the main parameters of the G formulation are required – for example over rock or desert sands. However, as is explained in section 3.2 "Refinements in soil heat flux parameterization", here we developed a new simple approach to estimate G based on a phase shift between LST and G to avoid errors using a constant fraction of net radiation over the diurnal cycle. The modifications derived here help to better capture thermal characteristics of the tundra substrate. Moreover, this method also investigates use of new scaling parameters that better reflect the thermal properties of the tundra soils, as noted by the reviewer.

RC_4: The authors give a mean value of 0.14 for cG and 0.92 for alphaPTC over the Arctic tundra. There is a rather a lot of handwaving here to suggest a single value for the entire Arctic tundra. What was the range of values across different vegetation types in the Arctic tundra. What was the error around the mean for this value? In addition what is the influence of changing cover over the growing season on both these values?

AC_4: A standard deviation has been included in the text for alpha and G values. AC_4: The initial values of the PTC use in this paper were the original value of 1.26 used in other TSEB applications and a value of 0.92 averaged from the main references found in the literature focused on Arctic tundra. As a starting point for the model we consider them applicable for areas of the Arctic with similar vegetation conditions. To clarify this within the text, the following paragraph has been added in section 2: "Under stress conditions, TSEB iteratively reduces PTC from its initial value. The TSEB model requires both a solution to the radiative temperature partitioning (Eq. 2) and the energy balance (Eqs. 6 and 7), with physically plausible model solutions for soil and vegetation temperatures and fluxes. Non-physical solutions, such as daytime condensation at the

soil surface (i.e., LES < 0), can be obtained under conditions of moisture deficiency. This happens because LEC is overestimated in these cases by the Priestley–Taylor parameterization, which describes potential transpiration. The higher LEC leads to a cooler TC and TS must be accordingly larger to satisfy Eq. (7). This drives HS high, and the residual LES from Eq. (11) goes negative. If this condition is encountered by the TSEB scheme, PTC is iteratively reduced until LES $\sim$ 0 (expected for a dry soil surface). However there are instances where the vegetation is not transpiring at the potential rate but is not stressed due to its adaption to water and climate conditions (Agam et al., 2010) or the fact that not all the vegetation is green or actively transpiring (Guzinski et al., 2013)."

RC_5: The use of MODIS LAI is particularly problematic in Arctic areas and it has been noted that the largest discrepancies in MODIS LAI are at Arctic tundra sites where the MODIS product overestimates woody cover proportions. Given that you have no LAI observations you cannot make any conclusions about how they relate to fPAR for example on page 13 line 30. What specific product was used, was it the 250 m resolution? What was the spatial extent of your footprint for this dataset and how does that relates to the spatial separation of your sites? Specifically which QC flags were used? How were gaps treated in the timeseries? Perhaps use MODIS fPAR. Given you have tower measurements of this you could validate the MODIS fPAR and assess the error here.

AC_5: The specific MODIS products used, and treatment of gap-filling and QC flags, are now more completely described in section 4.2.2.

RC_6: It is not clear as to how you distinguish between canopy and soil in these Arctic systems for the TSEB model. What do you define as soil and what is canopy? You have no significant woody vegetation to form a canopy in the first place. The surface layer consists of mosses, lichen, Forbes and shrubs and forms a continuous layer that cannot be partitioned into soil and canopy. I suspect in general you don't have any bare soil at your sites. Hence I'm not sure why you are using a two layer model

here in the first place? Can you justify the use of a two layer model here? Therefore the assumption that fPAR is equivalent to fG is not robust. To use this you will need to demonstrate clearly that this is the case. Do you even need a two layer model? Perhaps evaluate the usefulness of this type of model in this type of environment.

AC_6: The tundra canopy in the region where we have the tower measurements is dominated by a shrub canopy having an average height of 0.4 m. This overstory is likely to strongly affect the energy exchange and divergence of radiation and wind reaching the moss/lichen surface while the moss/lichen understory will act similar to a "bare soil" surface being aerodynamically smooth. The energy balance of the moss/lichen surface is computed using a 'bare soil" aerodynamic resistance for the sensible heat flux based on the "moss/lichen" temperature derived from Eq. (2), and with net radiation reaching this surface along with the estimated G term, the residual LE would then represent the mosses/lichen water use instead of bare soil. An assumption is that the soil resistance formulation is applicable to the moss/lichen understory. Given that the temperature partitioning derived from Eq. (2) which will yield a moss/lichen substrate temperature, significantly impacts the flux partitioning, using TSEB is assumed to be a reasonable approach for this ecosystem. AC_6: As it was explained at the beginning of the reviews, fG has been estimated using Guzinsky et al. (2013) methodology. This has been clarified in section 2.

RC_7: The description of the eddy covariance data is minimal. What software was used to process the data and what algorithms and parameters were used? Exactly what quality flags were filtered?

AC_7: The treatment of the EC data has now been expanded on in Section 4.2.3.

RC_8: What percentage of data were excluded due to different quality control previously mentioned as well as the three criteria mentioned.

AC_8: The first quality control excluded 20% of the data, accounting for inaccuracies in both meteorological and eddy covariance data. The second filter excluded 52% of the

data due to summer rainy conditions in the Arctic. After the precipitation filter, 10% of data was excluded because of a balance closure for 30 min timesteps less than 70%. Finally, to account for daily conditions (Rn > 100 WÂům-2 filter), around 50% of the remaining data was excluded.

RC_9: How were gaps in the data filled and worthy gap filled data used in the analysis?

AC_9: No gap filled data was used in this study; this was clarified in the text. Although gap filled data would have increase the final amount of data to evaluate the model, we preferred to have less data that are more reliable since they were derived from the measurements.

RC_10: The criteria of a surface energy balance closure of greater than 70% doesn't instill a lot of confidence in the measurements. I would assume from this that the energy balance closure is quite low. This is probably due to the difficulty in measuring the soil heat flux.

AC_10: As mentioned in section 6.3, "the average energy balance closure using half-hour periods for the evaluation dataset was 88% which is in agreement with the average closure of 90% for these flux stations, (Euskirchen et al., 2012)".

RC_11: The measures of performance are relatively standard so I don't think you need to include the formulas here but just cite a previous reference.

AC_11: We prefer to keep the formulas; we found that sometimes it is useful for the reader to have them in the text to better interpret the results.

RC_12: The distribution of residual energy based on the Bowen ratio is not a common practice and the community in general prefers to see the original data being used. This is overwhelmingly important in this environment where there are very large errors in measurements of G and also Rn, both of which go into the available energy term. Errors in these will propagate into errors in the turbulent heat flux terms if you force them based on the bon ratio. Calculating LE as the residual of the surface energy

balance equation is even more problematic as it is the sole term carrying all errors in the other terms. I would insist on redoing the analysis using only the original data and not presenting the other methods because they are so error prone.

AC_12: As explained in the text (section 6.3), lack of closure may be explained by instrument and methodological uncertainties, insufficient estimation of storage terms, unmeasured advective fluxes, landscape scale heterogeneity or instrument spatial representativeness, among others (Lund et al., 2014;Stoy et al., 2013;Foken et al., 2011;Foken, 2008;Wilson et al., 2002). Currently, there is no uniform answer on how to deal with non-closure of the energy balance in eddy covariance datasets, and methods for analyzing the reasons for the lack of closure are still under discussion (Foken et al., 2011). More recently there is evidence that non-orthogonal sonics underestimate vertical velocity causing under-measurement of H and LE on the order of 10% (Kochendorfer et al., 2012; Frank et al., 2013), although this is still being debated (Kochendorfer et al., 2013). This is the reason why in the current study a distribution of residual according to the Bowen ratio (BR) method was applied as suggested by Twine et al. (2000) and Foken (2008). In addition, LE was recalculated as the residual (RES) of the surface energy budget used in previous TSEB evaluations (Li et al., 2008). Foken et al. (2011) concluded that the different footprints of radiation, soil heat flux, and turbulent flux measurements, including the storage terms, which were postulated earlier to be a reason, have no significant influence on the energy balance closure results. In addition, the sonic anemometer and gas analyzer used in this study are Type A instrument have a typical accuracy between 5% and 10% for sensible and latent heat flux estimation, respectively while shortwave radiation and longwave radiation measure with the four components net radiometer have a 1% and 20 WÂům-2 accuracy (Foken, 2008). Additionally, the ground heat flux, including the storage term in the upper soil layer, can be determined with acceptable accuracy under most conditions (Foken, 2008). In our case we have a complete set of instrumentation to estimate G including soil bulk density data at each flux tower site.

RC_13: Table 2 shows the TCAV at 2 cm but this is usually an integrated measure with probes at two and 4 cm. Please check this.

AC_13: This has been clarified in the text. TCAV were placed in the soil at 2 and 4 cm depths.

RC_14: G is hard to measure. There is a great uncertainty in measurements of G in the tundra because traditional heat flux plates are made with an assumed thermal conductivity for loamy soils but we know in the tundra that this is primarily organic heat and moss which has a significantly lower thermal conductivity. Therefore self-calibrating heat flux plates or corrections are required. Can you quantify the uncertainty in your ground heat flux measurements which is an important term because it feeds directly into the energy balance?

AC_14: We have used self-calibrating soil heat flux plates. This has been clarified in Table 2. In addition, we have used the calorimetric method using soil bulk density data for each site to account for soil heat storage as it was explained in section 4.2 "Model inputs and evaluation datasets". This method has been also applied for Lund et al. (2014) for tundra conditions.

RC_15: How did you account for these in the correction of the soil heat flux plates? At what depth did you have the heat flux plates placed? I see they were 8 cm but is that below the surface in the moss? If so then your heat flux plates are not in soil but in organic material. You should use the appropriate bulk density not the soil bulk density. Also it appears that you only have one heat flux plate measurement per site which is insufficient given the spatial heterogeneity in the surface. As previously mentioned the thermal conductivity of the heat flux plate is manufactured to a standard soil which will not be representative of what you are measuring in. This will all result in very large errors in the observed soil heat flux.

AC_15: As explained above, we have used self-calibrating soil heat flux plates, TCAV water reflectometers to estimate G. All instruments are placed in the soil and not in

the moss layer. We have used the calorimetric method using soil bulk density data for each site to account for soil heat storage as it was explained in section 4.2 "Model inputs and evaluation datasets". This method has been also applied for Lund et al. (2014) for tundra conditions. The soil bulk density was already mentioned in the paper and it is 758 kgÂům-3, 989 kgÂům-3 and 1038 kgÂům-3 for Fen, Tussock and Heath flux stations, respectively.

AC_15: We agree with the reviewer that having more soil heat flux plates, TCAV and water reflectometers will improve the soil heat flux calculation. In table 2 we only listed the instruments but not the number of instruments per site. We have four self-calibrating soil heat flux plates, two water reflectometers and two thermocouple averaging soil temperature probes per flux station. This has been clarified in table 2. Similar instrumentation (same amount of instrumentation) is also used in many FLUXNET sites to address the spatial heterogeneity in the surface the soil.

RC_16: Please provide a thorough estimate of error and uncertainty for this particular important measurement. In addition, what is the uncertainty (random and model) in the fluxes for each of the sites?

AC_16: Soil heat flux model error is reported in detail under Section 6.1

RC_17: Given the difficulty in measuring G and the errors associated with that it may be worth trying to take G as a residual of the surface energy balance.

AC_17: In our case, G is a relatively small term compared with other fluxes, and as we explained before, lack of closure is likely to occur due to methodological uncertainties, insufficient estimation of storage terms, etc. when processing eddy covariance data (sensible and latent heat fluxes).

RC_18: As mentioned in the summary there is a lot of focus on model error and performance. However, these comparisons are with often in different types of models in different ecosystems which is like comparing apples and oranges. Most published

models will have some reasonable performance but we should move away from a simple reporting of the error to include better and more robust benchmarking of models. For example, this model could be compared against a simple empirical model to assess quantitatively whether the model performs any better than a simple model with local meteorological drivers. Recent papers have started to do and I suggest this is something that you could do to strengthen your paper. For example see:

Whitley, R., Beringer, J., Hutley, L., Abramowitz, G., De Kauwe, M. G., Duursma, R., Evans, B., Haverd, V., Li, L., Ryu, Y., Smith, B., Wang, Y.-P., Williams, M. and Yu, Q.: A model inter-comparison study to examine limiting factors in modelling Australian tropical savannas, Biogeosciences Discuss., 12(23), 18999–19041, doi:10.5194/bgd-12-18999-2015, 2015.

Luo, Y. Q., Randerson, J. T., Abramowitz, G., Bacour, C., Blyth, E., Carvalhais, N., Ciais, P., Dalmonech, D., Fisher, J. B., Fisher, R., Friedlingstein, P., Hibbard, K., Hoffman, F., Huntzinger, D., Jones, C. D., Koven, C., Lawrence, D., Li, D. J., Mahecha, M., Niu, S. L., Norby, R., Piao, S. L., Qi, X., Peylin, P., Prentice, I. C., Riley, W., Reichstein, M., Schwalm, C., Wang, Y. P., Xia, J. Y., Zaehle, S. and Zhou, X. H.: A framework for benchmarking land models, Biogeosciences, 9(10), 3857–3874, doi:10.5194/bg-9-3857-2012, 2012.

AC_18: Ultimately, this would be a goal for a follow-on paper. This paper focused on the utility of adapting/refining the TSEB land surface scheme for the Arctic tundra region represented by the flux tower sites used in this study. This is the reason we used Kalma et al. (2008) study as a robust benchmark for evaluating the performance of the TSEB relative to a large number of surface energy balance models using land surface temperature. In this paper, methods for estimating evaporation from landscapes, regions and larger geographic extents, with remotely sensed surface temperatures were reviewed, and uncertainties and limitations associated with those estimation methods were highlighted. In addition, particular attention was given to the validation of such approaches against ground based flux measurements. An assessment of some 30

published validations summarized in Kalma et al (2008) ranging from complex physical and analytical methods to empirical and statistical approaches) indicates a robust model should yield an average root mean square error (RMSE) value of around 50 W m-2 or less in estimated hourly turbulent fluxes H and LE during daytime conditions. The results from the current study yield RMSE values that fall generally below 50 W m-2 and hence considered a robust thermal-based energy balance model for the Arctic tundra.

RC_19: Page 14, line 3, the effect of what over the model? Mosses? In addition in this paragraph although you should not use the modus LA it is still consistent with seasonal growth of deciduous shrubs in particular. It is not inconsistent to have a constant fPAR where almost all incoming PAR is absorbed. The Arctic environment is highly adapted to absorbing as much energy as it can. As the leaf area of the shrubs increases during the summer the absorbed PAR is spread out amongst a greater leaf area but the fraction of fPAR remains the same.

AC_19: The lack of FAPAR consistency has been addressed in previous comments by using Guzinski et al. (2013) approach.

RC_20: Given this is a two layer model where are the results from the canopy and soil components.

AC_20: Although the TSEB model components the overstory and understory component fluxes, there are no measurements available to evaluate the reliability of the partitioning. This is a project planned for a future study when measurements of the component fluxes are available.

References

Asner, G. P.: Biophysical and Biochemical Sources of Variability in Canopy Reflectance, Remote Sens Environ, 64, 234–253, 1998.

Choi, M., Kustas, W.P., Anderson, M.C., Allen, R.G., Li, F., and Kjaersgaard, J.P., An intercomparison of three remote sensing-based surface energy balance algorithms over a corn and soybean production region (Iowa, U.S.) during SMACEX. Agric. Forest Meteorol. 149: 2082–2097. 2009. Euskirchen, E. S., Bret-Harte, M. S., Scott, G. J., Edgar, C., and Shaver, G. R.: Seasonal patterns of carbon dioxide and water fluxes in three representative tundra ecosystems in northern Alaska, Ecosphere, 3, art4, 10.1890/es11-00202.1, 2012. Foken, T.: The energy balance closure problem: an overview, Ecol Appl, 18, 1351-1367, doi:10.1890/06-0922.1, 2008. Foken, T., Aubinet, M., Finnigan, J. J., Leclerc, M. Y., Mauder, M., and U, K. T. P.: Results of a Panel Discussion About the Energy Balance Closure Correction for Trace Gases, B Am Meteorol Soc, 92, Es13-Es18, doi:10.1175/2011BAMS3130.1, 2011. Frank, J. M., Massman, W. J., and Ewers, B. E.: Underestimates of sensible heat flux due to vertical velocity measurement errors in non-orthogonal sonic anemometers, Agr Forest Meteorol, 171, 72-81, doi:10.1016/j.agrformet.2012.11.005, 2013. Guzinski, R., Anderson, M. C., Kustas, W. P., Nieto, H., and Sandholt, I.: Using a thermal-based two source energy balance model with time-differencing to estimate surface energy fluxes with day–night MODIS observations, Hydrol Earth Syst Sc, 17, 2809-2825, doi:10.5194/hess-17-2809-2013, 2013.

Huemmrich, K. F., Gamon, J. A., Tweedie, C. E., Oberbauer, S. F., Kinoshita, G., Houston, S., Kuchy, A., Hollister, R. D., Kwon, H., Mano, M., Harazono, Y., Webber, P. J., and Oechel, W. C.: Remote sensing of tundra gross ecosystem productivity and light use efficiency under varying temperature and moisture conditions, Remote Sens Environ, 114, 481-489, 10.1016/j.rse.2009.10.003, 2010.

Kalma, J. D., McVicar, T. R., and McCabe, M. F.: Estimating Land Surface Evaporation: A Review of Methods Using Remotely Sensed Surface Temperature Data, Surv Geophys, 29, 421-469, doi:10.1007/s10712-008-9037-z, 2008. Kochendorfer, J., Meyers, T. P., Frank, J., Massman, W. J., and Heuer, M. W.: How Well Can We Measure the Vertical Wind Speed? Implications for Fluxes of Energy and Mass, Bound-Lay Meteorol, 145, 383-398, doi:10.1007/s10546-012-9738-1, 2012. Kochendorfer, J., Meyers,

T. P., Frank, J. M., Massman, W. J., and Heuer, M. W.: Reply to the Comment by Mauder on "How Well Can We Measure the Vertical Wind Speed? Implications for Fluxes of Energy and Mass", Bound-Lay Meteorol, 147, 337-345, doi:10.1007/s10546-012-9792-8, 2012. Li, F. Q., Kustas, W. P., Prueger, J. H., Neale, C. M. U., and Jackson, T. J.: Utility of remote sensing-based two-source energy balance model under low- and high-vegetation cover conditions, J Hydrometeorol, 6, 878-891, doi:10.1175/Jhm464.1, 2005. Lund, M., Hansen, B. U., Pedersen, S. H., Stiegler, C., and Tamstorf, M. P.: Characteristics of summer-time energy exchange in a high Arctic tundra heath 2000-2010, Tellus B, 66, doi:10.3402/Tellusb.V66.21631, 2014. Norman, J. M., Kustas, W. P., Prueger, J. H., and Diak, G. R.: Surface flux estimation using radiometric temperature: A dual-temperature-difference method to minimize measurement errors, Water Resour Res, 36, 2263, doi:10.1029/2000wr900033, 2000. Tang., R., Li, Z-L., Jia, Y., Li, C., Sun, X., Kustas, W.P. and Anderson, M.C. An intercomparison of three remote sensing-based energy balance models using Large Aperature Scintillomter measurements over a wheat-corn production region. Remote Sensing of Environment. 115:3187-3202. 2011. Timmermans, W. J., Kustas, W. P., Anderson, M. C., and French, A. N.: An intercomparison of the Surface Energy Balance Algorithm for Land (SEBAL) and the Two-Source Energy Balance (TSEB) modeling schemes, Remote Sens Environ, 108, 369-384, 10.1016/j.rse.2006.11.028, 2007.

Twine, T. E., Kustas, W. P., Norman, J. M., Cook, D. R., Houser, P. R., Meyers, T. P., Prueger, J. H., Starks, P. J., and Wesely, M. L.: Correcting eddy-covariance flux underestimates over a grassland, Agr Forest Meteorol, 103, 279-300, doi:10.1016/S0168-1923(00)00123-4, 2000.

Stoy, P. C., Williams, M., Spadavecchia, L., Bell, R. A., Prieto-Blanco, A., Evans, J. G., and van Wijk, M. T.: Using Information Theory to Determine Optimum Pixel Size and Shape for Ecological Studies: Aggregating Land Surface Characteristics in Arctic Ecosystems, Ecosystems, 12, 574-589, 10.1007/s10021-009-9243-7, 2009. Wilson, K., Goldstein, A., Falge, E., Aubinet, M., Baldocchi, D., Berbigier, P., Bernhofer, C.,

Ceulemans, R., Dolman, H., Field, C., Grelle, A., Ibrom, A., Law, B. E., Kowalski, A., Meyers, T., Moncrieff, J., Monson, R., Oechel, W., Tenhunen, J., Williams, M., Rastetter, E. B., Shaver, G. R., Hobbie, J. E., Carpino, E., and Kwiatkowski, B. L.: Primary production of an arctic watershed: An uncertainty analysis, Ecol Appl, 11, doi:1800-1816, 10.1890/1051-0761, 2001.

---

## Author Response (AR2)

**AC:** We would like to thank the editor for his insightful comments and suggestions, which we believe have improved the quality of this manuscript.

**EC: Sentence 2 of the Introduction – can this reference be updated? The 2008 report is now referring to simulations from a decade ago**

**AC:** The reference has been updated.

**EC: Page 2Line 24 – leaf area index mentioned but not abbreviated; in the abstract LAI abbreviation is used but not defined .. please check use throughout of acronyms**

**AC:** This has been modified in the text.

**EC: Currently the paper sections 2-5 are methods and then section 6 serves as both results and discussion. It therefore remains difficult to link specific aims, approaches and findings through the manuscript. Currently at the end of Page 3 a general aim is given, and then some methodological/approach type statements. I believe this could be strengthened to better allow the reader to link specific objectives in the introduction to approach and then to findings. For example, ideally sub-headings in Section 6 would map to specific aims/objectives made clear at the end of the introduction. This ensures the reader can more easily navigate the different aspects of the paper from start to finish.**

**AC:** We have clarified the specific aims of the paper and sub-headings on section 6 have been adapted to these aims.

**EC: Section 4.2.2 – the abstract and introduction makes reference to remote sensing as a key component of the study, yet the sub-headings don't make clear where this aspect of the study is described. I assume it is this section 4.2.2 – can its sub-heading be made more clear/specific? Or perhaps an introductory paragraph in Section 4.2 to overview/summarise the range of data used to paint a clear picture early on?**

**AC:** Section 4.2.2 heading has been changed to : "4.2.2 Remote sensing input data: vegetation properties" to match it to the previous section heading: "4.2.1 Micrometeorological input data"

**EC: Section 5 – I think this section is currently not ideal as a major heading – can it instead be included in Section 3 or 4 with some minor changes to sub-heading titles? Also, because this section is attached at the end of the model and data sections, it**

**remains unclear what is being evaluated by these metrics. It is stated that "The performance of the TSEB model and possible refinements for Arctic tundra was evaluated …" … but at this point of the paper it is unclear what this means. Can you please re-word and better link this section to specific variables being modelled/measured, and how it is used to compare model approaches – this may be helped if it is incorporated in section 3 or 4.**

**AC:** Section 5 has been merged to section 4 and we have improved this section text accordingly. We detected a typo on Eq. 24 that has been corrected.

**EC: Section 6.3 – I have followed up with Reviewer 2 about RC_12, relating to closure of the energy balance. After consultation, I believe this section is still not satisfactory in making clear the issue of error accumulating in the residual term. Because we do not understand where the error is coming from (instrument, advection, footprint, radiation, soil heat flux, etc.) then it is not appropriate to try and force the energy balance to close. Calculating LE as a residual puts all of the error in the LE term which is unacceptable and could be misleading. Due to the uncertainty in the source of error and the fact that there may be biases in any particular term in the energy balance, it is not appropriate to force energy balance closure either. The conservative approach is to use the original data and not force to energy balance to close. This section should be updated and improved bearing this in mind.**

**AC:** We have modified this section comparing our results with the original LE eddy covariance values (without imposing balance closure); therefore, using the conservative approach suggested by the reviewer and the editor.

However, we note that closure corrections are common practice in the literature when comparing surface energy fluxes methods, and in comparison with unclosed comparisons, provide bounds on the range in probable model error. We, therefore, would like to retain discussion of the closure comparisons. These results facilitate comparisons between our findings and other existing studies where impact of closure has been evaluated.

**EC: Acknowledgements – Please consider acknowledging the help and inputs by the reviewers during the discussion process (should you believe it was helpful in improving the manuscript!)**

**AC:** We really think both reviewers have helped to improve the text. An acknowledgement has been included in this section.